# Wnt-Ror-Dvl signalling and the dystrophin complex organize planar-polarized membrane compartments in *C. elegans* muscles

Alice Peysson[1,3], Noura Zariohi[1,3], Marie Gendrel [2], Amandine Chambert-Loir[1], Noémie Frébault[1], Elise Cheynet[1], Olga Andrini [1] & Thomas Boulin [1] ✉

Cell polarity mechanisms allow the formation of specialized membrane domains with unique protein compositions, signalling properties, and functional characteristics. By analyzing the localization of potassium channels and proteins belonging to the dystrophin-associated protein complex, we reveal the existence of distinct planar-polarized membrane compartments at the surface of *C. elegans* muscle cells. We find that muscle polarity is controlled by a non-canonical Wnt signalling cascade involving the ligand EGL-20/Wnt, the receptor CAM-1/Ror, and the intracellular effector DSH-1/Dishevelled. Interestingly, classical planar cell polarity proteins are not required for this process. Using time-resolved protein degradation, we demonstrate that –while it is essentially in place by the end of embryogenesis– muscle polarity is a dynamic state, requiring continued presence of DSH-1 throughout post-embryonic life. Our results reveal the unsuspected complexity of the *C. elegans* muscle membrane and establish a genetically tractable model system to study cellular polarity and membrane compartmentalization in vivo.

Cell polarization is a fundamental process that organizes distinct subcellular domains within cells. The elaboration of polarized membrane sub-compartments allows these specialized regions to have unique protein compositions, signalling properties, and functional characteristics. Cellular compartmentalization mechanisms rely on a multitude of molecular pathways that engage the cytoskeleton, scaffolding proteins and signalling molecules. Within small cellular ensembles and larger tissues, cell-cell communication and external cues from secreted factors play a major role in regulating cell and tissue polarity.

Wnts are a class of secreted glycoproteins that play a pivotal role in cell polarity by transducing their signals through a variety of pathways. These Wnt pathways are conserved and critical in all metazoans.

There are two main classes of Wnt signalling pathways: the canonical Wnt/β-catenin cascade[1] and several β-catenin-independent, so-called non-canonical signalling pathways. Among these non-canonical pathways, the Wnt-Planar Cell Polarity (PCP) module plays a major role in organizing cells in the plane of a tissue. Indeed, PCP is important for many developmental processes regulating the formation of tissues and organs in animals. For instance, PCP proteins regulate the orientation of hair follicles and sensory cells, the shape and movement of cells during gastrulation (e.g., convergent-extension processes) and neural tube closure, or the positioning of cell extensions, such as cilia and axons[2–6].

These non-canonical Wnt signalling pathways are also conserved in the nematode *Caenorhabditis elegans*[7–11]. They have been shown

[1]Université Claude Bernard Lyon 1, CNRS UMR 5284, INSERM U1314, MeLiS, Lyon 69008, France. [2]Institut de Biologie de l'Ecole Normale Supérieure (IBENS), Ecole Normale Supérieure, CNRS, INSERM, Université Paris Sciences et Lettres Research University, Paris 75005, France. [3]These authors contributed equally: Alice Peysson, Noura Zariohi. ✉e-mail: thomas.boulin@univ-lyon1.fr

to control directional polarity processes, including convergent extension movements during gastrulation[12], migration of neuronal precursors[13,14], and the extension, guidance, and branching of neuronal projections[15–18]. Generally, antero-posterior (A-P) polarity decisions rely on short and long-range Wnt ligand gradients, which have been visualized in vivo[19,20]. In the early worm embryo, short-range signals between contacting cells provide positional instructive cues that direct polarized cell division by orienting the mitotic spindle[21]. Post-embryonically, Wnt signalling has been shown to control asymmetric cell divisions of epithelial seam cells and vulva precursor cells[9,22].

While PCP proteins have been shown to accumulate asymmetrically in some cases[12,23], molecular bridges formed by transmembrane proteins interacting in trans—a hallmark of PCP machineries in flies[3,24]—have however not been observed in *C. elegans*. PCP proteins, instead, act on an individual cell basis, and antero-posterior polarity decisions

rely on transient directional cues provided by PCP proteins and/or Wnt ligands[8,9,25]. Thus, despite the strong molecular conservation of Wnt signalling pathways, there is currently no clear example of planar cell polarity at the level of an entire tissue in *C. elegans*.

The musculature of *C. elegans* is organized into two ventral and two dorsal quadrants that run along the length of the body. Each quadrant comprises two bands of successive muscle cells[26]. These muscle cells and their sarcomeres attach to the epidermis through integrin-containing adhesive complexes, forming a network that covers the outer surface of the muscle cells. Additionally, the Dystrophin-Associated Protein Complex (DAPC) connects the intracellular actin cytoskeleton to the extracellular matrix along the muscle sarcolemma[26]. When viewed dorsally and ventrally, body wall muscles appear as diamond-shaped cells (Fig. 1A). They project membrane extensions, called "muscle arms", towards the neurites of presynaptic

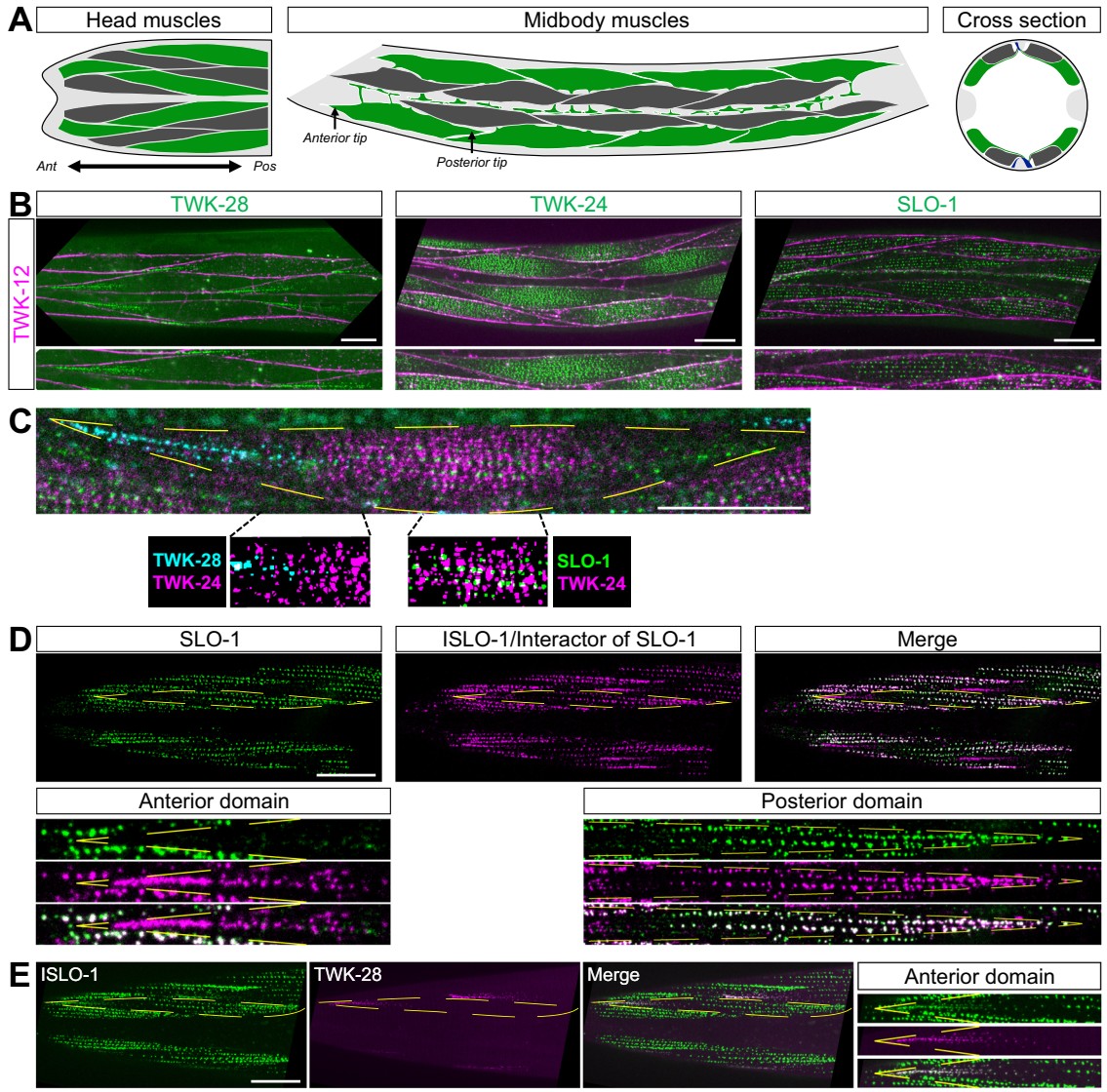

**Fig. 1 | Ion channel localization reveals polarized muscle membrane compartmentalization. A** Schematic diagram of the organization of the *C. elegans* musculature in the head, the midbody region, and in a cross section. Head and midbody dorso-ventral views show two out of four muscle quadrants. Successive muscle cells are labelled in green and dark grey. Anterior (*Ant*) to the left, posterior (*Pos*) to the right. Dorsal and ventral nerve cords are labelled in blue in the cross-section. Epidermal tissue is coloured in light grey. **B** Asymmetric localization of TWK-28, TWK-24 and SLO-1 channels. Muscle membrane labelled by TWK-12-wrmScarlet, magenta. **C** Image segmentation reveals little to no co-localization of ion channels

in triple-labelled muscle cells using TWK-28 (cyan), TWK-24 (magenta), and SLO-1 (green) fluorescent knock-in lines. **D** SLO-1 and ISLO-1 distributions coincide exclusively in the posterior domain. ISLO-1 is also found in the anterior domain of the muscle cell. **E** TWK-28 and ISLO-1 distributions overlap in the anterior domain of muscle cells. Rightmost column, magnified view of anterior domain of a muscle cell. Regions of interest in (**D**) and (**E**): *Anterior domain*, comet-shaped domain at anterior tip of muscle cells; *Posterior domain*, punctate clustered pattern in posterior part of muscle cells. Muscle cell outlines are indicated with yellow dashed lines. Scale bars, 20 μm.

motoneurons, establishing post-synaptic domains that concentrate ligand-gated ion channels at chemical synapses. The molecular mechanisms that underlie the construction of these membrane nanodomains have been extensively characterized[27]. In comparison, little is known about how membrane proteins are sorted to other parts of the muscle cell and whether they could be organized into distinct submembrane compartments. In fact, recent transcriptomic analyses have revealed that a surprisingly large number of ion channel genes are co-expressed in *C. elegans* muscle cells, raising interesting questions about their temporal dynamics, individual functions, and subcellular distribution[28–37].

In this study, we reveal the highly structured and polarized organization of the plasma membrane of *C. elegans* body wall muscle cells. By investigating the subcellular localization of ion channels and components of the dystrophin-associated protein complex, we demonstrate that these proteins localize to distinct and remarkably consistent asymmetric membrane compartments. Specifically, the potassium channels TWK-28, TWK-24, and SLO-1 occupy separate but partially overlapping polarized domains located at the tip, center, and posterior region of each muscle cell, respectively. DAPC proteins define further polarity patterns that differ in the anterior and the posterior of each muscle cell. The repetition of these asymmetric membrane compartments along the length of the body bears a striking resemblance to planar polarity patterns observed in other animals. However, we find that conserved PCP proteins are not required for this process. Through a candidate gene approach, we have found that muscle polarity relies on the Wnt ligand EGL-20, the Wnt receptor CAM-1/Ror, and the intracellular effector DSH-1/Dishevelled. Remarkably, DSH-1 is continuously required to maintain muscle polarity, and its post-embryonic re-expression can restore defective cell polarity, highlighting the dynamic nature of muscle polarity. Taken together, these findings uncover the intricate organization of the worm's sarcolemma and reveal a new instance of planar cell polarity in *C. elegans*.

## Results

### Ion channel localization reveals a remarkably complex and planar-polarized organization of the muscle sarcolemma

Potassium-selective channels are the largest family of ion channels in the *C. elegans* genome. Transcriptomic data indicate that up to 21 distinct potassium channel subunits are co-expressed in body wall muscle cells[28,34]. Yet, little is known about the subcellular localization of most of these channels. To address this question, we used two previously published reporter strains (TWK-18[32]; SLO-1[33]) and engineered knock-in lines for six two-pore domain (K2P) channels (TWK-8, TWK-12, TWK-24, TWK-28, TWK-42, TWK-43) using CRISPR/*Cas9*-based gene editing.

Careful analysis of these fluorescent reporters revealed a remarkable diversity of situations. While TWK-8 was present throughout the cell surface (Supplementary Fig. 1A), TWK-12 and TWK-43 were only visible in muscle arms and on the lateral sides of body wall muscle cells, but not on the outer surface that faces the hypodermis (Supplementary Fig. 1B). Conversely, TWK-18 and TWK-42 showed distinct distribution patterns on the outer face of each muscle cell, which were either broad (TWK-18) or punctate (TWK-42) (Supplementary Fig. 1A).

The localization of SLO-1, TWK-24 and TWK-28 were the most striking (Fig. 1A, B). Indeed, TWK-28 was concentrated at the anterior tip of each muscle cell, in a singular comet-shaped pattern. Conversely, SLO-1 channels were absent from the anterior part of the cell and formed a regularly-spaced punctate pattern in the centre and posterior region. Finally, while TWK-24 channels appeared to be broadly distributed at first glance, outlining muscle cells revealed that TWK-24 was in fact absent from their extremities and enriched in the central portion of the cell.

Given that SLO-1, TWK-24, and TWK-28 domains partially overlap at the scale of a muscle cell, we wondered whether they were also colocalized in membrane nanodomains. By simultaneously labelling the three channels, we observed that individual ion channel clusters were clearly separable using diffraction-limited confocal microscopy, suggesting that they could be part of distinct protein complexes, even where their distribution patterns overlap at the cellular scale (Fig. 1C).

The SLO-1 channel has been shown to physically interact with the integral membrane protein ISLO-1[36]. Therefore, we generated a fluorescent knock-in line to assess its subcellular localization. Remarkably, ISLO-1 and SLO-1 did not show identical distribution patterns. While ISLO-1 colocalized with SLO-1 in the posterior part of the cell (Fig. 1D), ISLO-1 was also clearly enriched at the anterior tip in a comet-shaped pattern that coincided perfectly with TWK-28 (Fig. 1E). Hence, the distribution pattern of ISLO-1 defines a fourth class of asymmetrical subcellular localization with both anterior and posterior domains, in which ISLO-1 is organized in very different subcellular patterns depending on its position in the cell, i.e., a comet-shaped anterior pattern, and a punctate, regularly-spaced, posterior pattern.

Taken together these observations reveal the remarkably complex molecular makeup of the plasma membrane of *C. elegans* muscle cells. They demonstrate the existence of asymmetric membrane compartments with distinct molecular compositions. Furthermore, as these asymmetric distribution patterns are repeated in each muscle cell all along the length of the worm, they also constitute the first examples of planar cell polarity at the tissue scale in *C. elegans*.

### Dystrophin and dystrophin-associated proteins are required for TWK-28 surface expression

To try to understand how the TWK-28 potassium channel is asymmetrically localized in muscle cells, we performed a forward genetic screen based on the rationale that the disruption of factors required to address TWK-28 to the cell surface would suppress the effect of a hyperactivating gain-of-function mutation. We previously demonstrated that the activity of vertebrate and invertebrate two-pore domain potassium channels can be tuned by mutating a key residue in the second transmembrane domain, named TM2.6[38]. After confirming that this mutation increased TWK-28 activity using heterologous expression in *Xenopus* oocytes (Supplementary Fig. 1C), we used CRISPR/*Cas9* gene editing to build a *twk-28* TM2.6 gain-of-function mutant (*bln485*, TWK-28 L210T). These mutants displayed strongly reduced locomotion, as expected for a gain-of-function mutation that decreases the excitability of muscle cells (Supplementary Fig. 1D).

After screening ~15.000 mutagenized haploid genomes, we obtained 182 independent suppressor lines that had regained near wild-type mobility. Among these, 102 were extragenic revertants based on genetic segregation, i.e., mutants that did not alter the *twk-28* gene sequence. Using whole genome resequencing, we analysed 25 of these mutants and found that 14 were loss-of-function alleles of *dys-1*, the *C. elegans* ortholog of dystrophin, and that one was an allele of *islo-1* (Supplementary Fig. 1E).

To measure their impact on TWK-28 surface expression, we combined *dys-1* and *islo-1* mutants with the fluorescently-labelled TWK-28 reporter (Fig. 2A, B). We observed a drastic change in *dys-1(bln582)* and *islo-1(bln549)* mutants as TWK-28 fluorescence was reduced by 74% and 70% relative to wild type, respectively. Interestingly, *dys-1(bln582)* harbours an early stop codon at W110. This mutation affects only the longest dystrophin isoform that contains both calponin-homology actin-binding domains. To verify whether the remaining TWK-28 fluorescence could be due to residual activity of shorter dystrophin isoforms, we generated *syb2174*, a molecular null allele that deletes the entire 31 kb-long *dys-1* genomic locus. This allele showed the same effect as *bln582*, arguing that the W110Stop mutation is a functional null allele for TWK-28 localization (Fig. 2A).

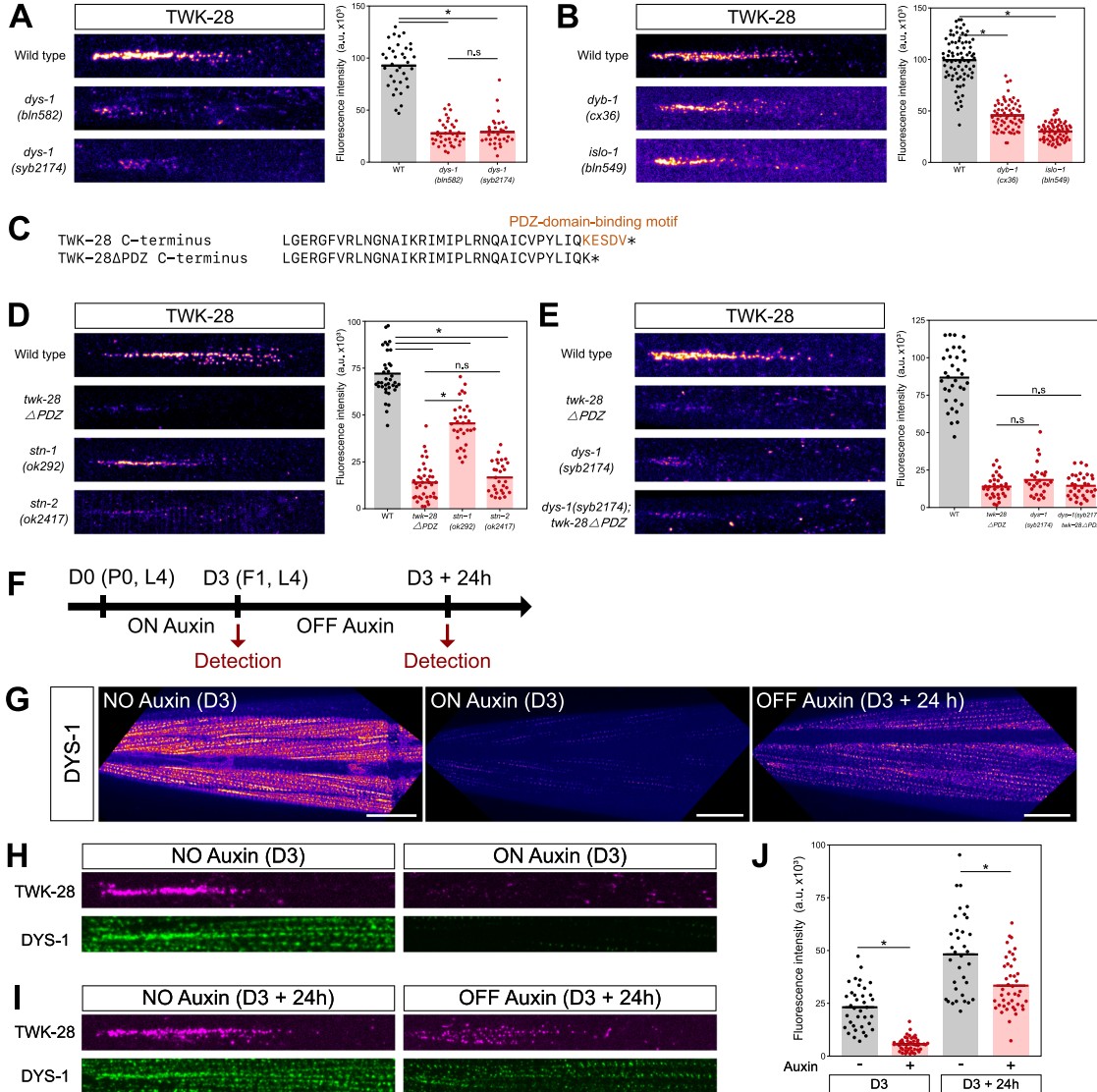

**Fig. 2 | Dystrophin and DAPC proteins determine TWK-28 content in the anterior domain of muscle cells. A** Confocal detection and quantification of mNeonGreen-TWK-28 in a long-isoform-specific *dys-1* mutant *(bln582)* and a *dys-1* molecular null allele *(syb2174)*. n = 36, 37, 31 (for respective columns), N = 3 independent experiments. **B** Confocal detection and quantification of mNeonGreen-TWK-28 in *dyb-1(cx36)* and *islo-1(bln549)* loss-of-function mutants. n = 78, 64, 71 (for respective columns), N = 3 independent experiments. **C** C-terminal sequence of the TWK-28ΔPDZ variant engineered by gene editing of mNeonGreen-TWK-28 to eliminate a putative PDZ-binding motif (orange residues). **D** Confocal detection and quantification of mNeonGreen-TWK-28 in *stn-1(ok292)* and *stn-2(ok2417)* loss-of-function mutants, and of an mNeonGreen-TWK-28 knock-in line lacking the final four C-terminal amino acids (*twk-28ΔPDZ*). n = 38, 39, 32, 29 (for respective columns), N = 3 independent experiments. **E** Confocal detection and quantification of mNeonGreen-TWK-28 and mNeonGreen-*twk-28ΔPDZ* in a *dys-1(syb2174)* mutant background. n = 35, 33, 27, 33 (for respective columns), N = 3 independent experiments. **F** DYS-1 degradation protocol. L4 (P0) worms are transferred to auxin-containing media at day 0 (D0). F1 progeny are exposed to auxin until the L4 stage (ON Auxin) and transferred to NGM plates (OFF Auxin) for 24 h. Confocal detection and quantification of fluorescence at day 3 (D3) and 24 h later (D3 + 24 h). **G** DYS-1-AID-mNeonGreen is strongly degraded after lifelong exposure to auxin (ON Auxin, D3). Partial recovery 24 h after removal from auxin (OFF Auxin, D3 + 24 h). Scale bars, 20 μm. **H, I** Confocal detection of wrmScarlet-TWK-28 and DYS-1-AID-mNeonGreen at D3 and D3 + 24 h, in the absence of auxin (NO Auxin, age-matched controls), after auxin exposure for 3 days (ON Auxin), and after removal from auxin for 24 h (OFF Auxin). **J** Quantification of wrmScarlet-TWK-28 fluorescence shown in (**H**) and (**I**). n = 36, 50, 34, 43 (for respective columns), N = 3 independent experiments. Kruskal-Wallis test with Dunn's multiple comparisons test; two-tailed Mann–Whitney test (for **J**). Each data point represents one muscle cell; bars represent data means; n.s., not significant, *p < 0.005. Raw data and statistical analyses are presented in the Source Data file. Image widths: 30 μm in **A, B, D, E**; 50 μm in (**H, I**).

In addition to dystrophin, *C. elegans* muscle express the orthologs of the dystrophin-associated proteins dystrobrevin (*dyb-1*), syntrophin (*stn-1, stn-2*), and sarcoglycan/Sgc (*sgca-1, sgcb-1, sgn-1*) (see below and Supplementary Table 1). We therefore tested the requirement of these DAPC proteins for TWK-28 localization using loss-of-function mutants. Individual sarcoglycan mutants and a triple knock-out did not modify TWK-28 fluorescence levels (Supplementary Fig. 1F and 1G). In contrast, *dyb-1(cx36), stn-1(ok292)* and *stn-2(ok2417)* each significantly

reduced TWK-28 fluorescence, by ~55%, 37%, and 76%, respectively, suggesting a predominant requirement of STN-2 (Fig. 2B, D). Importantly, in all cases, residual TWK-28 channels were still restricted to the anterior tip of muscle cells, indicating that targeting of TWK-28 to the comet-shaped domain is not solely dependent on DAPC proteins.

To assess whether dystrophin was more broadly required for the localization of muscle-expressed K2P channels, we combined four fluorescent knock-in lines with a *dys-1* loss-of-function allele. We

observed no obvious impact for any of these channels, whether they were uniformly distributed (TWK-8, TWK-18, TWK-42), or asymmetrically localized (TWK-24) (Supplementary Fig. 1A).

Taken together, these results suggested a model in which TWK-28 could be stabilized at the sarcolemma via interactions with the DAPC, as is the case for voltage-gated sodium channels and inwardly-rectifying potassium channels in vertebrate cardiomyocytes[39,40]. Indeed, a short amino acid sequence in the cytoplasmic C-terminus of $Na_v1.5$ and Kir4.1 is recognized by the PDZ domain of syntrophin, which itself binds to dystrophin and thus stabilizes these ion channels at the plasma membrane. Using in silico prediction algorithms[41], we identified a putative PDZ-binding sequence in the carboxy-terminus of TWK-28 (Fig. 2C). After inserting a premature stop codon that removed these amino acids by gene editing (Fig. 2C) in the context of the mNeonGreen-tagged TWK-28 fluorescent reporter strain, we observed a dramatic reduction in the number of TWK-28 channels (~80%) at the cell surface, similar to *dys-1* and *stn-2* mutants (Fig. 2D, E). Combining this truncated TWK-28 channel with a molecular null allele of *dys-1* did not further reduce TWK-28-associated fluorescence (Fig. 2E), which is consistent with a model in which TWK-28 is recognized by syntrophins and stabilized at the plasma membrane by the dystrophin-associated protein complex.

## TWK-28 dynamics following post-embryonic DYS-1 degradation and recovery

To examine the dynamics of the relationship between TWK-28 and the DAPC, we developed a temporally-controlled dystrophin degradation strategy using the auxin-inducible degron (AID) system[42]. To monitor DYS-1 levels and subcellular localization, we associated an mNeonGreen coding sequence with the AID degron sequence. This knock-in line was fully functional as we did not observe *dys-1* loss-of-function phenotypes (e.g., exaggerated head bending, reduction in TWK-28 surface expression).

Confocal imaging revealed a broad distribution of DYS-1 on the outer face of muscle cells, facing the epidermis (Fig. 2G). Consistent with the model that TWK-28 is recruited to the DAPC, DYS-1 and TWK-28 were well colocalized in the anterior tip of each muscle cell (Fig. 2H). Hence, while dystrophin is broadly distributed throughout the sarcolemma, it is also enriched at the tip of muscle cells.

We first tested whether DYS-1 could be degraded by observing the offspring of hermaphrodites grown on auxin (Fig. 2F, G). Although a very small fraction of the dystrophin-associated fluorescence remained detectable, it was drastically diminished in these auxin-treated worms (Fig. 2G, H). Consistently, worms exposed to auxin throughout their life displayed the exaggerated head-bending phenotype observed in dystrophin null mutants, confirming effective dystrophin degradation. Moreover, the impact of DYS-1 degradation was comparable to *dys-1* null mutants (Fig. 2A), since TWK-28 fluorescence was reduced by 73% compared to untreated animals (Fig. 2H, J).

Using this assay, we could analyse the kinetics of dystrophin recovery over 24 h, starting at the L4 larval stage (Fig. 2F). After removing animals from auxin-supplemented media, we saw a partial restoration of DYS-1-mNeonGreen fluorescence throughout the muscle cell and at the anterior tip. wrmScarlet-tagged TWK-28 channels also recovered notably over the 24 h time period, but did not fully reach wild-type levels (Fig. 2I, J). Notably, TWK-28 puncta were associated with DYS-1, in line with a direct association of TWK-28 with the DAPC (Fig. 2I). These data indicate that dystrophin complexes can be partially reassembled post-embryonically.

## Dystrophin-associated proteins localize to asymmetric membrane compartments

Given the polarized enrichment of dystrophin at the anterior tip of muscle cells, we used CRISPR/*Cas9* gene editing to label all members of

the worm's DAPC complex[43] and characterized their subcellular distribution.

In vertebrates, dystroglycan is a central component of the DAPC, linking the extracellular matrix to the intracellular cytoskeleton via dystrophin. In *C. elegans*, dystroglycan expression has been reported in different cell types, but surprisingly not in muscle[44]. However, expression of *dgn-1*, one of three worm dystroglycans, has been reported in body wall muscle by single-cell RNA sequencing approaches[28,34]. Consistently, DGN-1 was clearly visible in body wall muscles in our knock-in line, and DGN-1 was colocalized with DYS-1 on the outer face of muscle cells (Fig. 3A). Following these observations, we tested whether muscle DGN-1 also required the extracellular matrix protein perlecan[45], dystrophin[46] and dystrobrevin[47] for its surface expression as reported in vertebrates. Indeed, we could observe a profound disruption of DGN-1 surface expression in *dys-1 unc-52*/perlecan and mutants (Supplementary Fig. 2A, B). Loss of *dyb-1*/dystrobrevin alone had not effect, and a concomitant loss of *dys-1* and *dyb-1* had no additional effect compared to *dys-1* alone (Supplementary Fig. 2B). While the requirement of dystrobrevin may therefore not be conserved in worms, these results show a clear link between dystroglycan/DGN-1 and dystrophin/DYS-1 in *C. elegans* muscle.

To precisely analyse DGN-1 distribution, we used the dense body marker, PAT-2/α-integrin, as a cellular landmark[48]. Double labelling showed that both proteins occupy distinct membrane domains. Indeed, DGN-1 was excluded from dense bodies and M lines occupied by PAT-2 (Fig. 3B, C, D), even using diffraction-limited light microscopy. Dystrophin and dystroglycan are thus present throughout the sarcolemma of worm muscle cells, and occupy specific membrane nanodomains, juxtaposed to integrin-containing attachment sites.

Sarcoglycans/Sgc are the other major integral membrane component of the DAPC. The Sgc complex is composed of several sarcoglycan subunits (α, β, γ, δ) in skeletal and cardiac muscles[49]. The *C. elegans* genome encodes three sarcoglycans, *sgca-1*, *sgcb-1*, and *sgn-1*, which are orthologs of α-, β-, and δ/γ-sarcoglycan, respectively[43]. These three genes are co-expressed in worm muscles according to previous reports[36] and transcriptomic data[28]. Using knock-in lines, we confirmed their muscular expression and found that sarcoglycans are strictly co-localized (Supplementary Fig. 2C, D). As in vertebrates[49], sarcoglycan complex formation and membrane targeting was strongly impaired by the removal of individual subunits (Supplementary Fig. 2E), dystrobrevin or dystrophin (Supplementary Fig. 2F).

Interestingly, sarcoglycan distribution was visibly different from DYS-1, as it appeared more confined (Fig. 3E), and strongly resembled ISLO-1 (Fig. 1D, E). Indeed, sarcoglycans are also found at the anterior tip of the muscle cell in a comet-shaped pattern, and cluster into a linear pattern of regularly-spaced punctate microdomains in the posterior part of the cell. These two opposite domains are separated by a small region that appears to be mostly devoid of sarcoglycan (Fig. 3F, *gap domain*). By comparing Sgc and PAT-2 localization, we could observe that, similarly to dystroglycans, sarcoglycans occupy membrane domains that are clearly juxtaposed to integrin complexes (Fig. 3F). Remarkably, we could observe incomplete colocalization between DGN-1 and Sgc whereby some DGN-1 puncta were devoid of sarcoglycan—and vice-versa—(Fig. 3G), which suggests the existence of DAPCs with different protein compositions and demonstrates a further level of molecular complexity of the worm's sarcolemma.

Furthermore, we analysed the distribution of DYB-1/dystrobrevin, and the two worm syntrophins, STN-1 and STN-2. By comparing their subcellular distributions with DGN-1 and SGCB-1, we could see that they display two distinct localization patterns (Supplementary Fig. 3, 4). Indeed, STN-2 was broadly expressed and colocalized precisely with DGN-1, while STN-1 and DYB-1 reproduced the more confined distribution pattern of sarcoglycans. Finally, by using TWK-28 as a landmark, we confirmed that all DAPC proteins were colocalized with the channel at the anterior tip of muscle cells (Supplementary Fig. 5).

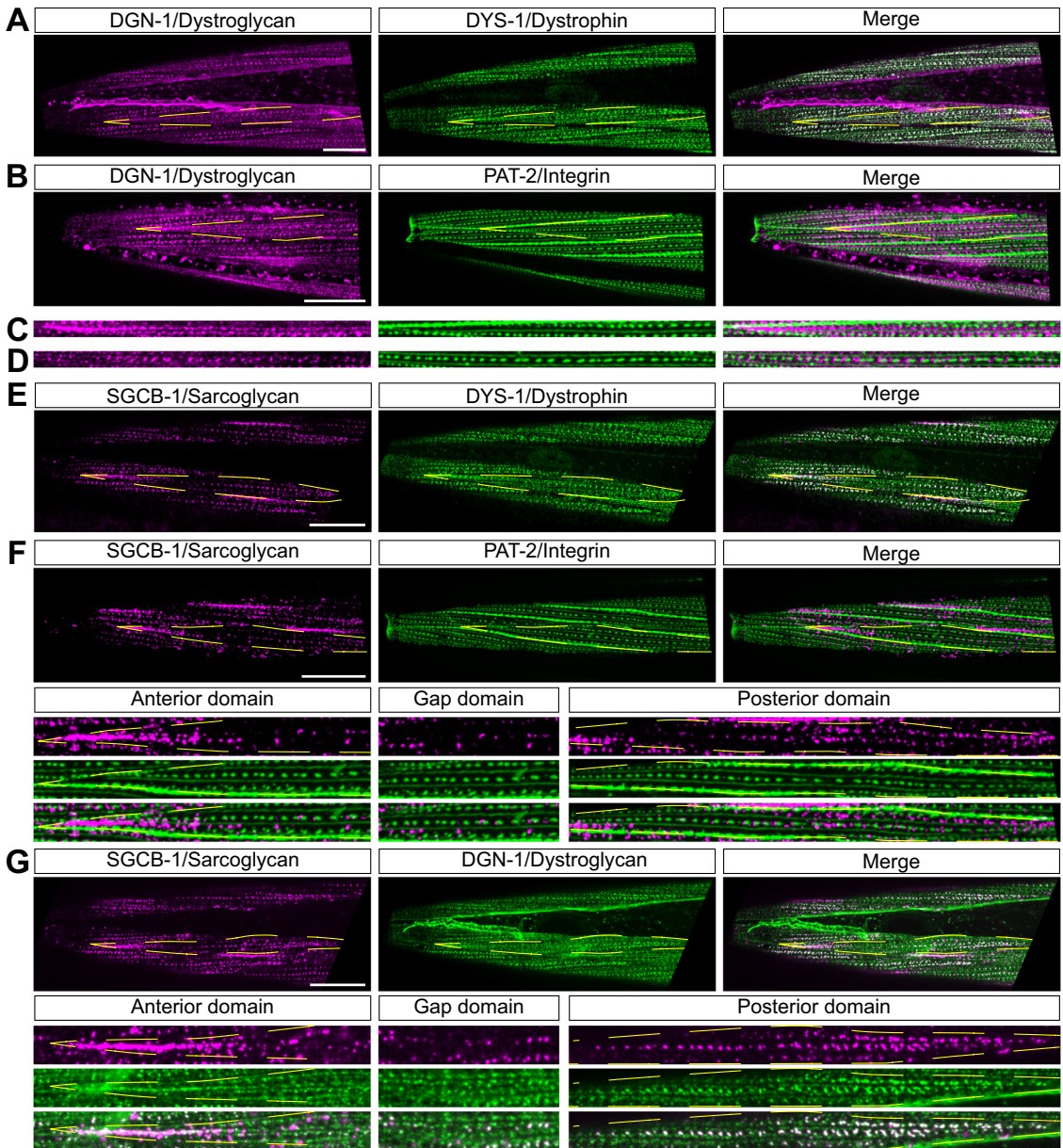

**Fig. 3 | Dystrophin, dystroglycan and sarcoglycans define asymmetrical membrane compartments at the surface of *C. elegans* muscle cells. A** DGN-1 and DYS-1 distributions overlap in body wall muscle cells. **B** DGN-1 localizes to membrane domains devoid of α-integrins/PAT-2. Representative images of head muscle cells of translational fusion knock-in lines for DGN-1-wrmScarlet (in magenta) and PAT-2-mNeonGreen (in green). **C**, **D** DGN-1 and PAT-2 occupy juxtaposed and optically separable membrane domains. Enlargement of images shown in panel **B**. **E** SGCB-1 and DYS-1 distributions overlap only partially in body wall muscle cells. **F** SGCB-1 partitions into distinct membrane domains along the antero-posterior axis of individual muscle cells. *Anterior domain*, comet-shaped domain. *Gap domain*, sparse localization. *Posterior domain*, clustered pattern aligned with, but optically separable from, PAT-2-labelled dense bodies. **G** DGN-1 and SGCB-1 patterns only partially coincide in the anterior and posterior regions of muscle cells. Muscle cell outlines are indicated with yellow dashed lines. Scale bars, 20 μm.

In conclusion, this analysis has revealed that DAPC proteins define multiple membrane subcompartments on the surface of muscle cells, distinct from integrin attachment sites. Moreover, the asymmetrical arrangements of these proteins in individual muscle cells create planar-polarized patterns throughout the muscle tissue.

### Dishevelled/DSH-1 controls the asymmetric distribution of membrane proteins

Our forward genetic screen revealed the requirement of DAPC proteins for stabilizing TWK-28 channels at the cell surface and lead us to uncover the polarized organization of the DAPC itself. Nevertheless, it did not elucidate the specific mechanisms governing muscle cell polarization.

Several molecular pathways are known to generate tissue polarity in different cellular contexts. Generally, these involve Wnt ligand/receptor systems and specific proteins belonging to the core planar cell polarity (PCP) pathway[3]. We hypothesized that the polarized organization of TWK-28 could also result from the activity of such tissue polarity pathways and undertook a candidate gene approach targeting Wnt and PCP genes that are conserved in *C. elegans*[9,25].

The intracellular effector dishevelled is a central player in these molecular cascades as it mediates both canonical and non-canonical Wnt signalling[50]. The *C. elegans* genome encodes three dishevelled orthologs: *dsh-1*, *dsh-2*, and *mig-5*. While mutation of *mig-5* or *dsh-2* had no defect (Supplementary Fig. 6A), when we combined a *dsh-1*

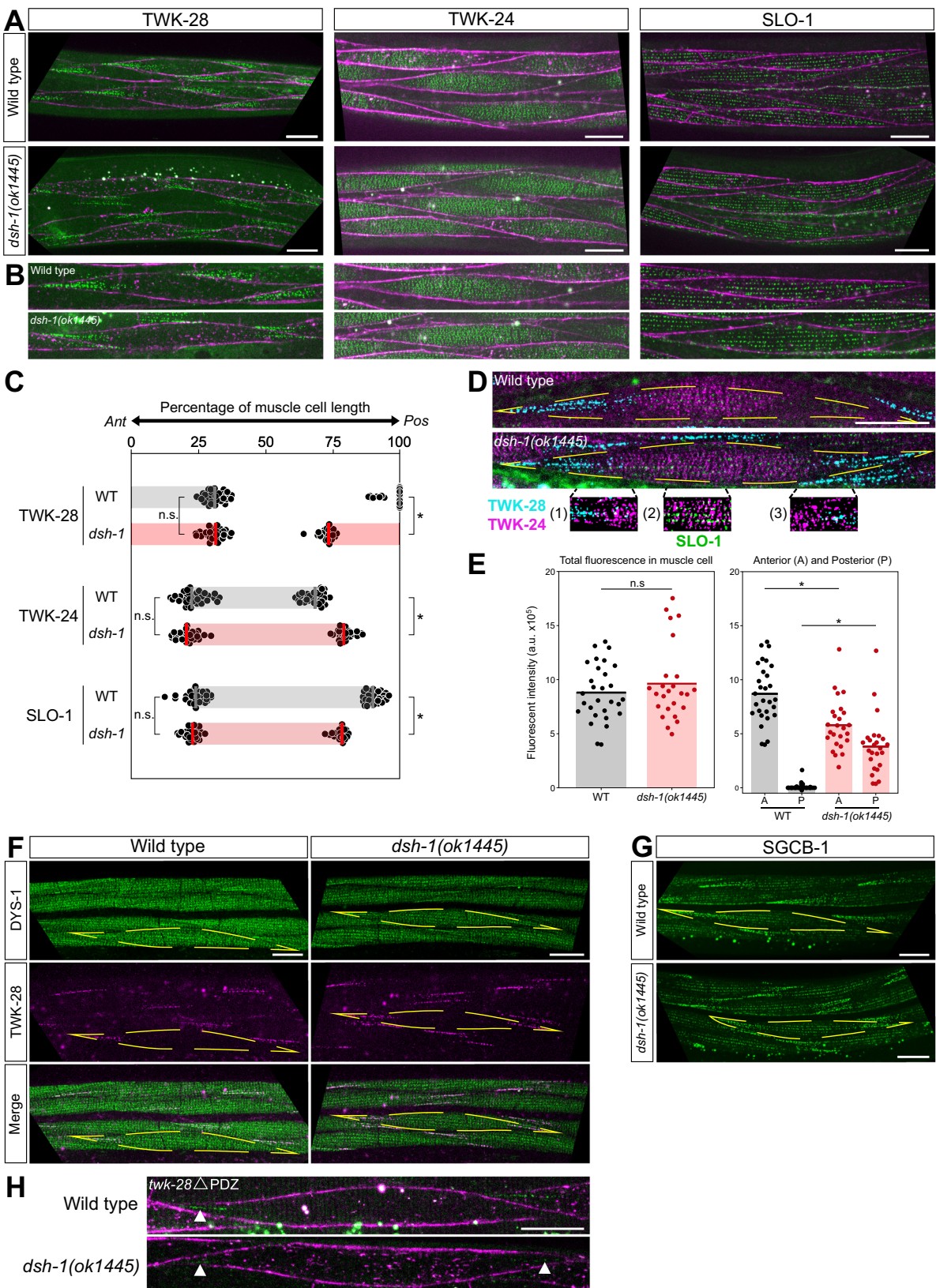

null mutant with the fluorescent SLO-1, TWK-24, and TWK-28 knock-in lines, we observed a striking reorganization of the channels at the muscle surface, leading to a loss of their asymmetric distribution (Fig. 4A, B). In particular, TWK-28 channels were now found at both extremities of the cell, giving rise to a head-to-tail configuration of anterior and posterior comets in adjacent muscle cells. SLO-1 and

TWK-24 distribution was also clearly modified as their domains shifted to a more central position in each cell.

To precisely analyze this redistribution, we measured the position −relative to the length of a given muscle cell−of the boundaries between membrane domains that included or that were devoid of channels (Fig. 4C).

**Fig. 4 | Loss of DSH-1/Dishevelled disrupts the planar-polarized organization of the muscle plasma membrane. A, B** Symmetrical distribution of TWK-28, SLO-1 and TWK-24 in a *dsh-1(ok1445)* loss-of-function background. Muscle membrane labelled by TWK-12-wrmScarlet (in magenta). **B** Magnified view of single muscle cells in wild type and *dsh-1(ok1445)*. **C** Quantification of TWK-28, SLO-1 and TWK-24 domain boundary positions in wild type and *dsh-1(ok1445)*. Average position of anterior (*Ant*) boundaries in wild type (black) and *dsh-1(ok1445)* (red), respectively, as percentages of muscle cell length: TWK-28, 31% and 30%; SLO-1, 24% and 23%; TWK-24, 23% and 21%. Average position of posterior (*Pos*) boundaries in wild type and *dsh-1(ok1445)*, respectively, as percentages of muscle cell length: TWK-28, 97% and 73%; SLO-1, 90% and 78%; TWK-24, 68% and 80%. Grey and red bars indicate the region containing each ion channel. Number of cells assayed in each condition: TWK-28 (WT, *n* = 29; *dsh-1*, *n* = 25); SLO-1 (WT, *n* = 30; *dsh-1*, *n* = 26); TWK-24 (WT,

*n* = 33; *dsh-1*, *n* = 26). *N* = 3 independent experiments. Bars represent data means. Two-tailed Mann–Whitney test, n.s. not significant, *$p$ < 0.0001. **D** Image segmentation reveals little to no co-localization of ion channels in *dsh-1*-mutant muscle cell using TWK-28 (cyan), TWK-24 (magenta), and SLO-1 (green) fluorescent knock-in lines. **E** Total TWK-28 amount is unchanged in *dsh-1(ok1445)* as TWK-28 is redistributed between the anterior and posterior domains. *n* = 29, 25, 29, 29, 25, 25 (for respective columns), *N* = 3 independent experiments. Two-tailed Mann–Whitney test, n.s. not significant, *$p$ < 0.0001. **F** Symmetrical distribution of DYS-1 and TWK-28 at extremities of muscle cells in *dsh-1(ok1445)*. **G** Symmetrical distribution of SGCB-1 in *dsh-1(ok1445)*. **H** Symmetrical distribution of TWK-28 lacking the PDZ binding-motif in *dsh-1(ok1445)*. White arrowheads indicate remaining fluorescent signal. Muscle cell outlines are indicated with yellow dashed lines. Scale bars, 20 μm. Raw data and statistical analyses are presented in the Source Data file.

First, we found that the anterior border was unchanged in *dsh-1(0)* mutants in each case. TWK-28 channels covered on average 30% of the anterior portion of the cell, while SLO-1 and TWK-24 boundaries were more anterior (23% and 21% respectively) and therefore overlapped slightly with the TWK-28 domain.

In contrast, the posterior borders underwent significant displacement. In wild type, sporadic clusters of TWK-28 channels could sometimes be observed at the cell's posterior end, while in *dsh-1(0)* mutants, TWK-28 channels covered over 25% of this region. Thus, anterior and posterior TWK-28 domains were now almost symmetrical. As for SLO-1, the posterior boundary that generally extended almost to the end of the cell, now shifted anteriorly, uncovering close to a fourth of the posterior portion of the cell. Conversely, the TWK-24 domain extended further posteriorly in *dsh-1(0)*, closely aligning with the distribution of SLO-1. Image segmentation of triple-labelled cells demonstrated that even in diffraction-limited conditions, the three channels remained optically separable. This suggests that they still form physically distinct nano-domains within the sarcolemma when muscle polarity is disrupted (Fig. 4D).

Next, we wondered whether the redistribution of TWK-28 channels also affected their surface expression levels. Through in vivo fluorescence measurements, we determined that the overall quantity remained constant in *dsh-1* mutants. Indeed, TWK-28 fluorescence was now divided between anterior and posterior comets, aligning with the size of each domain (Fig. 4E). These data thus suggest that the number of TWK-28 channels present at the cell surface is limited and finely regulated by mechanisms that operate independently of dishevelled/ *dsh-1*.

Considering the necessity of dystrophin for TWK-28 surface expression, we postulated that the distribution of DYS-1 would also be modified in *dsh-1* mutants. Using TWK-28 as a landmark, we could clearly observe that dystrophin was indeed present at both extremities of muscle cells (Fig. 4F) in a *dsh-1* null mutant. Likewise, the sarcoglycan SGCB-1 displayed a symmetrical distribution pattern, further reinforcing the idea that *dsh-1* plays a major role in ensuring the polarity of muscle cells (Fig. 4G).

Lastly, to further investigate the interplay between processes that control TWK-28 surface distribution via dystrophin and DSH-1, we used the TWK-28 mutant lacking the C-terminal PDZ-binding sequence (Fig. 2C), as it decouples TWK-28 from the DAPC. In this context, we could detect TWK-28 channels at the anterior and posterior ends in *dsh-1* null mutants (Fig. 4H), arguing that TWK-28 channel are addressed to the extremities of muscles cells independently of their interaction with the DAPC, and that the DAPC likely serves to stabilize TWK-28 at the surface in a second step.

Taken together, these data support a model in which DSH-1 ensures that dystrophin and TWK-28 are not addressed or maintained at the posterior end of muscle cells, while the DAPC, subsequently and independently, stabilizes TWK-28 channels at the cell surface. The process that addresses DYS-1, SGCB-1, and TWK-28 to the extremities of muscle cells remains to be determined.

### The DIX domain of DSH-1 is dispensable for TWK-28 asymmetry

Dishevelled proteins consist of three conserved functional domains: DIX, PDZ and DEP (Fig. 5A). Generally, DIX and PDZ domains are required for canonical Wnt β-catenin signalling, whereas PDZ and DEP domains function in PCP[50]. To investigate which pathways may act downstream of DSH-1 in muscle cells, we expressed a series of protein truncations in body wall muscle cells and monitored the rescue of TWK-28 localization in a *dsh-1* mutant background (Fig. 5B).

First, we verified that a *dsh-1a* cDNA was sufficient to restore TWK-28 localization (i.e., panel "+ *Pmyo-3::dsh-1a*"). Next, we expressed truncated *dsh-1a* cDNAs lacking each domain using the same muscle-specific promoter, and found that only the cDNA lacking the DIX domain restored TWK-28 localization in a *dsh-1* null mutant. In line with the absence of a DIX domain requirement, the distribution of TWK-28 remained unaffected in a *bar-1*/β-catenin mutant (Supplementary Fig. 6B).

Taken together, these results show that DSH-1 likely does not engage canonical β-catenin-dependent signalling in this context and point to an involvement of a PCP-type pathway.

### Sarcolemmal asymmetry requires Wnt/Ror but not PCP proteins

Based on the findings from our DSH-1 structure-function analysis, we examined the potential participation of genes associated with two planar polarity pathways: the core PCP proteins, VANG-1/Stan, FMI-1/ Flamingo, and PRKL-1/Prickle, and the global proteins, CDH-1/Dachsous and CDH-3/CDH-4/Fat. Surprisingly, none of these genes were required for planar-polarized TWK-28 localization (Supplementary Fig. 6B).

Given these results, we investigated the involvement of Wnt-dependent pathways. First, we tested a viable *mig-14*/Wntless hypomorphic mutant that reduces the secretion of Wnt ligands and found that it led to a loss of TWK-28 polarity (Fig. 5D). Next, we analyzed viable mutants for four Wnt ligands. We found that only *egl-20* altered the localization of TWK-28 (Fig. 5C and Supplementary Fig. 6C). Finally, we tested five transmembrane proteins that bind Wnt ligands: LIN-17, MIG-1, CFZ-2, LIN-18/Ryk and the receptor tyrosine kinase CAM-1/Ror. *lin-17* and *lin-18*, alone, and *mig-1 cfz-2* double mutants had no effect (Supplementary Fig. 6C, D). However, *cam-1* mutants phenocopied *egl-20* (Fig. 5C, E). To confirm these results, we generated an early stop codon at position Pro21 in EGL-20 using CRISPR/*Cas9* gene editing (Supplementary Fig. 6C) and analysed three additional *cam-1* alleles (*ak37, cw82, ks52*) (Fig. 5D, E). These four additional mutant alleles all showed phenotypes identical to the reference alleles.

Observing the impact of these gene mutations in more detail along the entire length of the animal provided some additional information about the molecular actors upstream of dishevelled. Specifically, *egl-20* and *cam-1* mutants only affected the distribution of TWK-28 in muscle cells located posteriorly to the vulva. In contrast, *dsh-1* and *mig-14* mutation also impacted muscle cells situated more anteriorly (Fig. 5C).

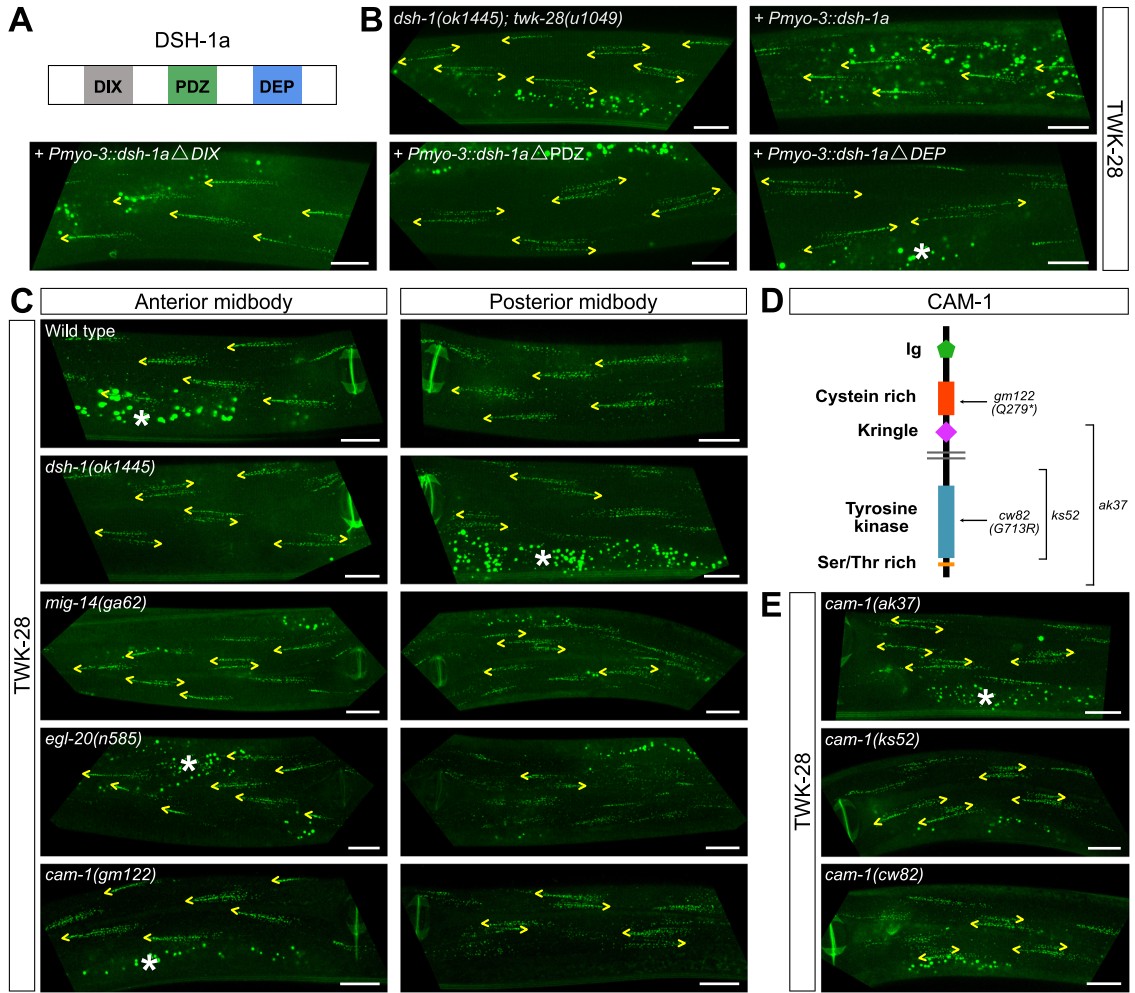

**Fig. 5 | Loss of MIG-14/Wntless, EGL-20/Wnt and CAM-1/Ror disrupts asymmetric localization of TWK-28. A** Schematic protein structure of DSH-1a including DIX, PDZ, and DEP domains. **B** Muscle-specific expression of full-length or DIX domain-truncated DSH-1a proteins rescues TWK-28 localization in *dsh-1(ok1445)*. PDZ and DEP domains are necessary for DSH-1 function in muscle polarity. **C** Loss of MIG-14/Wntless, EGL-20/Wnt, and CAM-1/Ror disrupts polarized localization of TWK-28. Posterior mid-body muscle cells are affected in all mutant genotypes. In *cam-1* and *egl-20* mutants, polarity is not affected in muscle cells situated anteriorly to the vulva ("Anterior midbody"). **D** Schematic protein structure of CAM-1/Ror receptor, with corresponding molecular lesions in point mutants (*gm122* and *cw82*), and deletion alleles (*ak37* and *ksS2*). **E** *cam-1(ak37), cam-1(ks52)*, and *cam-1(cw82)* disrupt TWK-28 polarity in posterior mid-body muscles. Left- or right-pointing yellow arrowheads indicate anterior or posterior extremity of muscle cells, respectively. Non-specific intestinal lysosome-related auto-fluorescence is visible in some panels and labelled with a white asterisk. Scale bars, 20 µm.

This spatially-restricted effect is consistent with the known range of the EGL-20 gradient that is formed by secretion from cells in the tail of the worm, extending to the midbody[19,20]. Furthermore, loss of CAM-1 also did not affect cells anterior to the vulva, which could indicate that a different Wnt receptor/ligand combination is acting in these cells, or that redundant pathways control muscle polarity in the anterior half of the worm.

### DSH-1 is asymmetrically localized in body wall muscle cells
To gain further insight into how DSH-1 controls muscle polarity, we analysed the subcellular distribution of DSH-1 in muscle cells using a translational knock-in lines in which the mNeonGreen coding sequence was inserted at the C-terminus (*bab365*). Remarkably, DSH-1 was enriched in the posterior third of each body wall muscle cell (Fig. 6A, B), clearly distinct of TWK-28's anterior localization (Fig. 6C). This pattern was lost in *cam-1* and *egl-20* mutants (Supplementary Fig. 7A) suggesting a link between DSH-1 localization and EGL-20/CAM-1 signalling. Surprisingly, while *egl-20* disrupted DSH-1 localization only in muscle cells that were posterior to the vulva—consistent with the EGL-20 gradient—loss of *cam-1* appeared to affect DSH-1

distribution all along the body. Yet, TWK-28 asymmetry was not affected in *cam-1* mutants anterior to the vulva, while loss of *dsh-1* altered TWK-28 polarization in these cells. Further experiments will be necessary to reconcile these observations.

### DSH-1 is continuously required to establish and maintain muscle membrane polarity
Next, we wondered whether muscle polarity was a fixed or a dynamic state, i.e., whether DSH-1 was only required to establish muscle polarity initially, or also to maintain it throughout the life to the worm. Therefore, in addition to mNeonGreen, we inserted the AID degron sequence into our DSH-1-mNeonGreen knock-in line to be able to manipulate DSH-1 protein levels and study its temporal window of action.

First, to validate this approach, we exposed fourth stage (L4) larvae to auxin and—3 days later—analysed their F1 progeny at the L4 stage. Using a transgenic line expressing the TIR1 ubiquitin ligase ubiquitously[51], we could observe a robust degradation of DSH-1-AID-mNeonGreen and a clear redistribution of TWK-28 channels into a symmetrical pattern, phenocopying a *dsh-1* null mutant (Supplementary Fig. 7B).

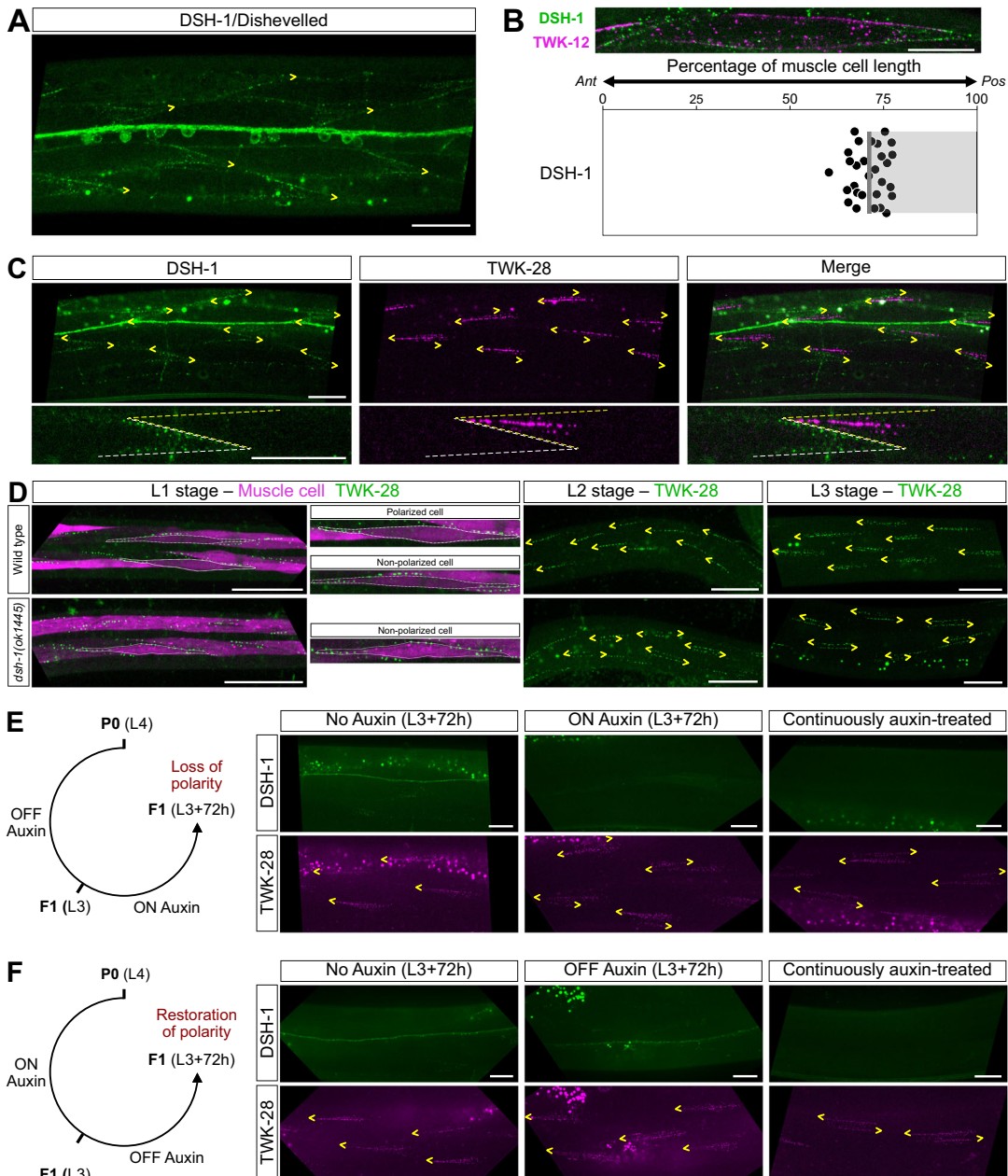

**Fig. 6 | DSH-1/Dishevelled is required to establish and maintain muscle polarity. A** Representative micrograph of an mNeonGreen translational knock-in line of DSH-1. DSH-1 is enriched in the posterior portion of body wall muscle cells. DSH-1-mNeonGreen is visible in ventral nerve cord motoneuron cell bodies and neurites. Ventral view. **B** DSH-1-mNeonGreen distribution in a single muscle cell. Muscle membrane labelled by TWK-12-wrmScarlet (in magenta). Average position of DSH-1 domain boundary, 72 % of muscle cell length, $n = 29$. Each data point represents one muscle cell. $N = 3$ independent experiments. *Ant*, anterior; *Pos*, posterior. Grey bar indicates the region containing DSH-1. **C** DSH-1 and TWK-28 are found at opposite ends of muscle cells. Dashed lines indicate the outline of muscle cell. **D** Asymmetric localization of TWK-28 in L1, L2, and L3 larval stages. Non-polarized muscle cells are found in wild type at the L1 stage. Loss of *dsh-1* disrupts muscle polarity at all stages. Muscle cells are labelled with cytoplasmic mCherry in the L1 stage and outlined with white dotted lines. **E** DSH-1 is required to maintain muscle polarity during post-embryonic development. **F** Muscle polarity is restored by re-expression of DSH-1 during post-embryonic development. Left- or right-pointing yellow arrowheads indicate anterior or posterior extremity of muscle cells, respectively. Scale bars, 20 μm. Raw data are presented in the Source Data file.

Next, to determine the appropriate time window for DSH-1 degradation, we observed the localization pattern of TWK-28 during the first three larval stages, in wild-type and *dsh-1(0)* worms (Fig. 6D). While muscle polarity was fully established at the second (L2) and third (L3) larval stages, we observed both polarized and non-polarized cells in freshly-hatched (L1) larvae. Notably, this suggests that TWK-28 polarity is mostly established during embryogenesis, and that it likely derives from an initially non-polarized state.

We also observed a progressive change in the shape of TWK-28-containing comets. Only a small number of fluorescent clusters were observed in the L1. However, in the L2 and L3 stages, the number of clusters increased, accompanied by a widening of the comet over time. Loss of *dsh-1* disrupted muscle polarity at all larval stages (Fig. 6D).

To determine whether *dsh-1* was required to maintain muscle polarity once it was established, we initiated the degradation of DSH-1-mNeonGreen at the L3 larval stage and examined these worms 72 h later. This protocol led to a clear loss of TWK-28 asymmetry,

underscoring the dynamic nature of muscle polarity and the continuous requirement of DSH-1 for its maintenance (Fig. 6E).

Conversely, we asked whether reinitiating DSH-1 expression post-embryonically could restore muscle polarity, beginning from a non-polarized state. In this case, we exposed animals to auxin until the L3 stage, and monitored the recovery of DSH-1 levels and muscle polarity over 72 h in the absence of auxin (Fig. 6F). As for L4 stage animals (Supplementary Fig. 7B), L3 animals exposed to auxin throughout their life showed a clear loss of muscle polarity and an absence of DSH-1-associated fluorescence. In turn, over the course of the 72 h recovery period, we observed a restoration of the polarized distribution pattern of TWK-28.

These two complementary experiments directly demonstrate that DSH-1 is required –throughout the life of the animal – to establish and to maintain the polarity of *C. elegans* muscle cells. This demonstrates that muscle polarity is a dynamic state and that it could be regulated by extracellular signals that impact DSH-1 activity.

## Discussion

By analyzing the subcellular distribution of potassium channels and proteins linked to the dystrophin-associated protein complex (DAPC), we reveal here the planar-polarized organization of *C. elegans* muscle cells. The diverse distribution patterns of these proteins underscore the complex makeup of the worm's sarcolemma. Despite decades of detailed work on the development, structure and function of *C. elegans* muscle cells[26], this remarkable instance of cellular polarization was entirely unsuspected.

We have shown here that muscle polarity is controlled by the Wnt ligand EGL-20, the Wnt receptor CAM-1/Ror and DSH-1/Dishevelled. Signalling cascades involving these proteins have been shown to regulate neurite outgrowth[52], neuroblast migration[13], acetylcholine receptor dynamics at neuromuscular junctions[53], and contribute to asymmetric cell division in epithelial cells[54]. However, the downstream effectors of these cascades remain largely unknown in *C. elegans*.

Among the different non-canonical Wnt pathways in vertebrates, the homologous Wnt5a-ROR-Dvl pathway has garnered significant attention. Perturbation of this signalling cascade in different animal models leads to tissue morphogenesis defects during embryogenesis. In humans, mutations in Wnt5a, ROR2, and Dishevelled proteins cause brachydactyly type B (BDB) and Robinow syndrome. Also known as Robinow dwarfism or foetal face syndrome, it is a rare genetic disorder with significant morphogenetic abnormalities that primarily affect the development of the bones and other parts of the body[4]. Genetic experiments in mice have demonstrated that loss of Wnt5a is phenocopied by concomitant loss of ROR1 and ROR2 receptors[55]. This study also demonstrated that Wnt5a signalling via Ror receptors leads to Dishevelled phosphorylation in vivo. Interestingly, CAM-1 is a known interactor of DSH-1 in worms[52], and we have shown here that *cam-1(ks52)* mutants, which lack the intracellular kinase domain, phenocopy *cam-1* null mutants, suggesting that kinase activity may also be required for muscle polarization, possibly through DSH-1 phosphorylation.

Furthermore, proteomic studies have revealed that Wnt5a-Ror signalling modulates the steady-state protein levels of two downstream targets, the kinesin Kif26b and the LNX family member Pdzrn3. Wnt5a-dependent phosphorylation of these proteins triggers their proteasomal degradation. Degradation of Kif26b alters the migratory behaviour of mesenchymal cells in culture[56]. Similarly, Kif26b regulates endothelial cell polarity by reorganizing and stabilizing the microtubule network to achieve directional cell growth in the developing vascular system[57]. In the retinal vasculature, the E3 ubiquitin ligase Pdzrn3 is required for the coordinated and directional extension of endothelial cell[58]. The abundance of Pdzrn3 is also dynamically regulated by the Wnt5a-Ror cascade, and the phosphorylation of three amino acids within the highly conserved LNX3H domain is required for this process[59].

Consistent with the notion that the Wnt-Ror-Dvl cascade may dynamically control intracellular effectors, a remarkable feature of *C. elegans* muscle polarity is that it is a plastic state. Indeed, *C. elegans* muscle polarity remains visible throughout the life of the cell, much like the sensory organs built by PCP processes in fly epithelia and in the inner ear of vertebrates. However, in contrast to these actin-based structures, membrane proteins are generally subjected to constant turnover. This may explain why the continuous activity of DSH-1 is required to maintain membrane organization and muscle polarity. It will, therefore, be essential to identify the molecular targets of DSH-1 in muscle cells and to identify additional factors required to establish and maintain muscle polarity. Forward genetic screens and candidate gene approaches will be necessary in the future to elucidate the cellular mechanisms that establish and maintain muscle polarity in *C. elegans*.

Given the essential role of non-canonical Wnt5a-Ror-Dvl signalling in development and human disease, the mechanisms by which this signalling cascade regulates cell biological functions are still insufficiently understood. Therefore, delineating the molecular actors controlling muscle polarity in *C. elegans* may shed light on the conserved mechanisms and molecular effectors that act downstream of Wnt-Ror non-canonical signalling cascades.

PCP proteins are generally distributed asymmetrically within polarized cell, which provides a directional cue to break cell symmetry and establish planar cell polarity. As shown here, muscle polarity does not appear to require PCP proteins – at least not when they are disrupted individually. How could symmetry breakage be achieved in this context?

Asymmetric localization of DSH-1 is a frequent feature of polarization processes in different species and cellular contexts. For example, Dishevelled is enriched distally in sensory hair cells of the mammalian organ of Corti[60] and in *Drosophila* wing epithelial cells[61]. Asymmetric distributions of Dishevelled have also been described in RME neurons[62] or between neurites of the bipolar PLM neuron[17] in *C. elegans*. We have shown here that DSH-1 displays a clear asymmetric localization in muscle cells. Based on genetic epistasis tests, this posterior enrichment depends on EGL-20/Wnt and CAM-1/Ror. DSH-1 asymmetry could therefore result from a graded activation of CAM-1 by the EGL-20 gradient. Interestingly, in the developing mouse limb, the Wnt5a receptor ROR2 does not display an asymmetric distribution in chondrocytes[63], but it has been proposed that its polarizing function results from a gradient of activity. Indeed, an instructive gradient of Wnt5a activity in the chondrocyte field is translated into a gradient of Vangl2 activity via the formation of the ROR2 and Vangl2 protein complex. A model has therefore been proposed in which ROR2 captures the Wnt5a concentration gradient with a dose-dependent response, resulting in Vangl2 phosphorylation according to the Wnt5a gradient.

Based on these observations, it is possible that the posterior-to-anterior oriented EGL-20/Wnt gradient could differentially activate CAM-1/Ror at the posterior end of each muscle cell, and subsequently recruit and activate DSH-1 asymmetrically. Such a model implies that EGL-20 may have an instructive role in establishing the polarity of muscle cells. To test this prediction, it would be interesting to express EGL-20 uniformly and assess whether the gradient provides an instructive or a permissive signal. For instance, it has been shown in this way that the ground polarity of vulval precursors cells is established by the instructive activity of EGL-20 via CAM-1 and VANG-1[22].

Recent observations have revealed intriguing asymmetric distribution patterns and functions of the DAPC in other cellular contexts. Precise in vivo studies of dystrophin in mouse models have been greatly improved by the generation of GFP-tagged mouse knock-in lines. These models have confirmed the broad distribution of dystrophin along the membrane of muscle fibres[64,65] but also revealed a

striking enrichment at the extremities of muscle fibres where the functions of dystrophin remain to be fully determined[66].

Asymmetric localization of dystrophin has also been described in zebrafish muscles[67]. Analysis of a gene trap line that fluorescently labels the endogenous dystrophin protein showed that dystrophin is highly enriched at myosepta –the contact site that separate two somites– forming a repetitive chevron-like structure along the length of the body of the fish. During development, more dystrophin is recruited to this interface, exacerbating the asymmetric organization of dystrophin within growing muscle fibres.

In activated rodent muscle stem cells (i.e., satellite cells) dystrophin is located apically in close contact with the extracellular matrix. This polarized distribution of dystrophin regulates the asymmetric division of satellite cells via the recruitment of the serine-threonine kinase Par1b[68].

Finally, in *Drosophila*, dystrophin is polarized in epithelial migrating cells, where it is found in the posterior half of cells in a complex pattern, illuminating both the cell cortex and filamentous structures partially aligned with F-actin stress fibres[69]. Functional studies have found that the DAPC promotes planar polarization of integrin clusters and participates in the trafficking of ECM components leading to the formation of polarized fibrils[70].

Taken together, these observations and our findings suggest that the DAPC may be at the heart of cellular mechanisms that precisely define subcellular cortical domains, allowing specific functions to be compartmentalized within a single cell, and extending its role beyond that of a mere structural component.

Although DAPC proteins are often thought to form a single molecular complex, the situation is more complicated at mammalian neuromuscular junctions where they are all present but not colocalized and not involved in the same functions[71]. Our observations also challenge the notion of a singular DAPC composition.

At first glance, DAPC components can be categorized into two groups: one comprising DGN-1, DYS-1, and STN-2, and the other encompassing sarcoglycans, DYB-1, and STN-1. One-to-one colocalization revealed that there is significant, but not complete overlap between both groups. For example, the posterior sarcoglycan puncta coincide with areas of denser DGN-1 signal, which is surrounded by weaker DGN-1-associated fluorescence (Fig. 3G). This could be explained by different diffusion kinetics with one predominant population of confined sarcoglycans *versus* two populations of confined *versus* freely diffusing dystroglycans. Moreover, a heterogeneous organization of dystroglycan- and sarcoglycan-containing complexes is visible in the anterior tip of muscle cells where some DGN-1 puncta are devoid of sarcoglycan, and vice-versa.

To definitively clarify DAPC localization in worm muscles and determine the exact molecular makeup of dystrophin-containing nanodomains, it will be essential to transition from diffraction-limited microscopy to super-resolution techniques, as the substantial increase in resolution will clarify the composition of membrane nanodomains and spatial distribution of dystrophin and its associated proteins. Indeed, even when proteins appear co-localized using conventional confocal microscopy, they could still occupy distinct nanodomains. Refining their relative location will reveal the heterogeneity within DAPC complexes at the surface of muscle cells. For example, the two worm syntrophins, STN-1 and STN-2, have distinct and partially overlapping distribution patterns in muscle cells. We have also found that they contribute differently to the surface expression of the potassium channel TWK-28. Fly and vertebrates also co-express multiple syntrophins with different subcellular localizations and functions[72]. Thus, understanding the interplay between these two worm syntrophins may provide new leads on the mode of action of syntrophins. In the future, one could employ fluorescence recovery or photoconversion techniques to explore the dynamics of various DAPC components in vivo. This approach may unveil both molecular and functional diversity within the complex.

Finally, it is also interesting to note the clear separation of DAPC and integrin complexes in worm muscles. Using diffraction-limited microscopy, we could easily distinguish these complexes suggesting that they are situated far from each other. These observations refine previous analyses that suggested a close association with dense bodies in the I-band[73]. It thus remains to be determined which molecular partners or scaffolds ensure the highly stereotyped subcellular distribution of DAPC proteins.

Whether and how ion channels are distributed to different domains within muscle cells has been rarely investigated specifically, except for the extensive work regarding the targeting of ligand-gated ion channels to neuromuscular junctions[27]. Based on our analysis of single-cell RNAseq data and other published reports, up to 21 potassium channel subunits are likely co-expressed in body wall muscle cells. By comparing the localization of seven of these, we have uncovered a diversity of situations. For instance, TWK-8 and TWK-18 channels are found throughout the sarcolemma, while TWK-12 and TWK-43 channels are only targeted to the basolateral membrane and muscle arms. Conversely, TWK-28 and TWK-42 channels are only found on the outer side of the muscle cell, facing the epidermis, and not on the lateral membrane or muscle arms. Importantly, the comparison of TWK-28, TWK-24, and SLO-1 channels shows that even when channels are located in the same region of the sarcolemma (i.e., the outer face of the cell), they can occupy distinct –yet partially overlapping– regions along the antero-posterior axis. And even when they are present in overlapping regions, ion channels remain confined to distinct nanodomains. The remarkable consistency of these patterns across various individuals and within the muscle cells of a single animal suggests the existence of precise molecular and cellular mechanisms responsible for directing individual channels to their designated domains, likely endowing them with distinct electrophysiological properties.

Another remarkable observation is the difference in organization of TWK-28, ISLO-1 and SLO-1 (Fig. 1). ISLO-1's interaction with SLO-1 and the syntrophin STN-1 recruits the BK potassium channel to the DAPC. In turn, mutations in *islo-1* and *dys-1* have been demonstrated to disrupt the surface expression of SLO-1 within muscle cells[36]. Yet, we have shown here that ISLO-1 and SLO-1 localization patterns are not identical since ISLO-1 is also present in the comet-shaped domain at the anterior tip of each muscle cell. Similarly, we have found that *islo-1, dys-1* and syntrophins are necessary for TWK-28 surface expression. Yet, TWK-28 is restricted to the anterior domain. Notably, even when cellular polarity is disrupted by mutations in *egl-20*/Wnt, *cam-1*/Ror and *dsh-1*/Dishevelled, the posteriorly-localized TWK-28 does not reproduce the posterior, punctate, ISLO-1 pattern, but rather mirrors the comet-shaped distribution seen at the anterior tip in wild-type animals. Therefore, there are likely other factors that allow the selective recruitment of TWK-28 and SLO-1 to distinct anterior and posterior domains. An important question in the future will be to understand the functional implications of these specific ion channel distributions on the control of muscle excitability and contraction.

## Methods
### *C. elegans* methods and genetics
All *C. elegans* strains were originally derived from the wild-type Bristol N2 strain. Worm cultivation, genetic crosses, chemical mutagenesis and manipulation of *C. elegans* were carried out according to standard protocols[74]. All strains were maintained at 20 °C on nematode growth medium (NGM) agar plates with *Escherichia coli* OP50 as a food source. Animals used for phenotyping are hermaphrodites. Strains and alleles used for this study are listed in Supplementary Data 1 and 2, respectively.

## Molecular biology

Single-strand oligonucleotides, crRNA, and plasmids used in this study are described in Supplementary Data 3, 4, and 5, respectively.

## Auxin-induced degradation

Auxin plates were prepared by adding auxin Indole-3-Acetic Acid (Sigma) from a 400 mM stock solution in ethanol into NGM at the final concentration of 1 mM[42]. DYS-1 and DSH-1 degradation was performed using a transgene expressing TIR1 with the ubiquitous promoter *Peft-3*[51]. Protein degradation was monitored in vivo based on mNeonGreen fluorescence.

## General microscopy methods

Freely moving worms were observed on nematode growth media (NGM) plates using an AZ100 macroscope (Nikon) equipped with a Flash 4.0 CMOS camera (Hamamatsu Photonics). For confocal imaging, animals were mounted on 2% dry agarose pads with 2 % polystyrene beads (Tebu-Bio, polystyrene polybeads 0.10 μm microsphere) in M9 buffer (3 g $KH_2PO_4$, 6 g $Na_2HPO_4$, 5 g NaCl, 0.25 g $MgSO_4 \cdot 7 H_2O$, and distilled water up to 1 litre). Confocal imaging was performed using an inverted confocal microscope (Olympus IX83) equipped with a confocal scanner unit spinning-disk scan head (Yokogawa) and an EMCDD camera (iXon Ultra 888, Andor) at the Ciqle imaging facility (Centre d'Imagerie Quantitative Lyon-Est, LyMIC-CIQLE, Lyon, France).

## TWK-28-mNeonGreen fluorescence intensity quantification

Quantification of fluorescent images was performed using ImageJ (version: 2.0.-rc-69/1.53a). Each data point represents one muscle cell and data from three independent imaging sessions were pooled for each genotype. Acquisition parameters were the same across genotypes for quantitative analyses. Fluorescence intensity was measured for Fig. 2A to E in a region of interest of 30 μm (long) x 5 μm (wide) x 2.5 μm (deep), and for Fig. 2J in a region of interest of 50 μm (long) x 5 μm (wide) x 2.5 μm (deep). To quantify the fluorescence intensity of TWK-28 in Fig. 4E, we segmented individual muscle cells using TWK-12::wrmScarlet to outline the muscle membrane. The total fluorescence intensity was corrected by subtracting background fluorescence from the middle of the cell, where TWK-28 was undetectable.

## Measurement of TWK-28, TWK-24 and SLO-1 domain boundaries

To determine the precise outline of individual muscle cells, we used TWK-12::wrmScarlet for cell membrane segmentation. We then measured the length of each cell and the relative position of the TWK-28, TWK-24 and SLO-1 domain boundaries along the antero-posterior axis using ImageJ (Fig. 4C).

## Electrophysiology and heterologous expression of TWK-28 in *Xenopus laevis* oocytes

Capped mRNAs were synthesized in vitro from linearized expression vectors using the T7 mMessage mMachine kit (Ambion, Austin, TX, USA). Defolliculated *X. laevis* oocytes (Ecocyte Bioscience, Dortmund, Germany) were injected with 50 nL containing 1.8 ng of cRNA. Oocytes were kept at 18 °C in ORII Calcium solution containing (in millimolar): 82.5 NaCl, 2 KCl, 1 $MgCl_2$, 0.7 $CaCl_2$, 5 HEPES, gentamicin (25 μg/mL), pH 7.5 (with TRIZMA-Base).

Two-electrode voltage clamp (TEVC) experiments were performed 24 h after microinjection. Oocytes were mounted in a small home-made recording chamber and continuously superfused with ND96 solution containing (in millimolar): 96 NaCl, 2 KCl, 1.8 $CaCl_2$, 2 $MgCl_2$, 5 HEPES. pH 7.4 was adjusted with Trizma base. Macroscopic currents were recorded using a Warner Instrument OC-725 amplifier, filtered at 10 kHz, digitized using a Digidata-1322 (Axon Instrument). For current visualization and stimulation protocol application, we used Axon pClamp 9 software (Molecular Devices, Sunnyvale, CA).

Recording electrodes were pulled to 0.2–1.0 MΩ by using a horizontal puller (Sutter Instrument, Model P-97, USA) and filled with 3 M KCl. Currents were recorded in response to a voltage-step protocol consisting of a pre-pulse of −80 mV (80 ms duration) from a holding potential of −60 mV, followed by 10 mV steps (200 ms duration) from −150 mV to +50 mV, and return to a −60 mV holding potential. Current-voltage curves were obtained by plotting the steady-state currents at the end of each voltage step.

## Statistics and reproducibility

Statistical analyses were performed using Prism (version 10.2.3) and are reported in the Source Data file. Graphs were plotted using Prism or the PlotsOfData service[75]. In *dsh-1, mig-14, egl-20* and *cam-1* mutant animals, polarity was assessed in >25 animals for each genotype, and scored as defective when more than four cells were found to be symmetrical. Conversely, we considered that a given *dsh-1* cDNA was functionally sufficient to rescue muscle polarity if the *dsh-1* polarity defect was reversed in more than four cells of more than ten transgenic animals. Representative micrographs were selected among a random sampling of more than thirty individuals.

## Reporting summary

Further information on research design is available in the Nature Portfolio Reporting Summary linked to this article.

# Data availability

Worm strains and plasmids generated in this study are available upon request. The raw data generated in this study are provided in the Source Data file. Further information and requests for resources and reagents should be directed to and will be fulfilled by Thomas Boulin. Source data are provided with this paper.

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

## Acknowledgements
We thank Hannes Bülow, Jean-Louis Bessereau, Vincent Mirouse, Bénédicte Durand and Martin Harterink for critical reading and helpful comments about the manuscript. We are grateful to Terese Lawry and Martin Chalfie for sharing *C. elegans* strains. We thank Driss Laabid and Sandra Duperrier for technical assistance. This work was funded by grants to T.B. from the European Research Council (ERC Stg 2013, *Kelegans*), AFM Téléthon (Alliance MyoNeurALP), and ANR (AAPG 2023, *DYSCO*). A.P. was supported by AFM Téléthon and LabEx CORTEX. We thank *Le Centre d'Imagerie Quantitative Lyon-Est* (LyMIC-CIQLE, Lyon, France) imaging facility for support and access to equipment. Some strains were generated by SEGiCel (SFR Santé Lyon Est CNRS UAR 3453, Lyon, France). Some strains were provided by the CGC, which is funded by NIH Office of Research Infrastructure Programs (P40 OD010440). Some strains were provided by the National Bioresource Project at the Tokyo Women's Medical University School of Medicine, which is part of the International *C. elegans* Gene Knockout Consortium.

## Author contributions
A.P., N.Z., M.G., A.C.-L., N.F., and E.C. performed experiments in *C. elegans*. O.A. performed two-electrode voltage-clamp experiments. All authors were involved in the analysis of the experimental data. T.B. supervised the work. A. P. and T. B. wrote the manuscript.

## Competing interests
The authors declare no competing interests.
