## [Peer Review File · Nature Communications]

REVIEWER COMMENTS

Reviewer #1 (Remarks to the Author):

This is a comprehensive and very interesting study. Using the *C. elegans* muscle cells as a model, the authors showed that a subgroup of potassium channels (TWK-28, TWK-24, and SLO) are asymmetrically localized within individual muscle cells, and that this asymmetry is extended across the entire musculature, similar to classical planar cell polarity (PCP) in the fly and vertebrate epithelia. This observation is novel, as such tissue-level planar polarity has not been previously described in *C. elegans*. Using an extended series of genetic analyses, the authors went on to demonstrate that members of the dystrophin-associated protein complex (DAPC) play an important role in membrane expression/retention of the potassium channels, but not in these channels' asymmetrical distribution. Importantly, the classical PCP pathway also appears to be dispensable for asymmetrical channel distribution, and instead the Wnt-Ror-Dvl pathway was found to be required. The study presents an elegant example where the Wnt-Ror-Dvl pathway imparts asymmetry at both the cellular and tissue levels, and that this occurs independently of the classical PCP pathway. It is likely that this functional distinction between the two morphogenetic signaling systems, as revealed by the study, is conserved in other biological contexts, and in other metazoan species. The experimental paradigm described in the study should also provide a powerful tool for future mechanistic investigation of Wnt-Ror-Dvl signaling in symmetry breaking. Overall, I am enthusiastic about the work and believe that it represents a significant advancement in the field.

Suggestions to authors:

1. The phenotypes described in Figs. 5B-D, 5F, and 6E, 6F should be quantified and statistically analyzed. Ideally, statistical analysis should also be done for the *dsh1* effects on asymmetrical localization of TWK-28, SLO-1 and TWK-24 shown in Fig. 4A-C.
2. Presentation of Fig. 5 can be improved. The boundaries of individual muscle cells are difficult to discern without the magenta *Twk-12-wrmScarlet* marker as shown in Fig. 1 and 4. In the legend, the authors stated "Left- or right-pointing yellow arrowheads indicate anterior or posterior extremity of muscle cells, respectively", however, it is still difficult to understand what these arrowheads are supposed to show, since many panels only have left pointing arrowheads, in the middle of the green *twk-28* signals. Showing zoomed-in images of the muscle cells with more clearly marked boundaries would be helpful. It would also be helpful to annotate the green signals as *twk-28* (as shown in Fig. 1B and 4A).
3. In Fig. 5D legend, the authors stated "Anterior mid-body muscle cells in *cam-1* and *egl-20* mutants are wild type". By "wild type", do they mean "phenotypically normal"? This should be made more clear.

4. There appear to be many more of the very large punctate green twk-28 signals (e.g. Panel D, wild type, anterior midbody, near the left bottom quadrant of the image; Panel D, dsh-1 posterior midbody, bottom part of the image). Is there an explanation for this, and what might these large puncta be?

5. Fig. 6A, again the use of the arrowheads is confusing. It is also difficult to distinguish the white and yellow arrowheads in the rather small images.

Reviewer #2 (Remarks to the Author):

In this study, Peysonn et al. investigate the localization of potassium channels and the dystrophin-associated protein complex in muscle cells in the nematode *C. elegans*. Unexpectedly, the authors uncover highly specific subcellular localization patterns for different potassium channels. Moreover, the localization patterns appear to be governed by some type of planar cell polarity mechanism, with channels localizing to the anterior, posterior, or center of muscle cells.

In the next part of the manuscript the authors investigate the requirement of different components of the dystrophin-associated protein complex (DPAC) for localization of the TWK-28 channel, and the subcellular localization of DPAC components.

In the final part of the manuscript the authors investigate the mechanism behind the planar polarization of the channels and the DPAC. They find that the Wnt-Ror-Dvl pathway is necessary for the planar polarization of muscle cells.

The identification of planar polarized localization patterns for ion channels in muscle is both exciting and novel, as to my knowledge muscle cells in *C. elegans* were not previously known to show planar polarity. I found the overall study to be impressively complete and well documented, with extensive investigations into the entire DPAC 'complex', the ion channels, potential PCP pathways, Wnt ligands and other Wnt signaling components. Many new transgenic lines were made to investigate subcellular localization and co-localizations, structure function analyses are presented of DSH-1 and CAM-1, and developmental requirements are investigated using inducible protein degradation.

Given the conserved nature of ion channels, DPAC components, and Wnt signaling, the findings are likely to be relevant to a broad audience beyond the *C. elegans* community. I gladly recommend publication, and have only a few small points that might be addressed.

1. Keeping track of the many components tested for non-muscle specialists would be helped by a schematic overview of the possible DPAC components and their mammalian/worm names.

2. The final sentence of the introduction ‘first examples of quintessential planar cell polarity patterns’ is overstating the findings. Canonical PCP pathways are not involved in muscle, and planar Wnt polarity is well established in the seam cells. The study is exciting enough without claiming to be the first ever to show PCP in worms.
3. In line 217 I was expecting Figure S1A to show the ‘no obvious impact’, but it only shows the wild-type distribution of channels. I don’t think the lack of effect of *dys-1* loss is shown. It would be more appropriate to list this as data not shown.
4. Line 254: analyse dystrophin dynamics. This to me implies looking at protein dynamics (e.g. FRAP), but it is expression/translation dynamics that are followed.
5. On line 333: three Dsh orthologs. *Mig-5* and *dsh-1* are investigated but *dsh-2* is left out. Is there a specific reason for not mentioning *dsh-2* here? The rest of the study is so complete, it seems an odd omission.
6. I did not understand the analysis in Figure 4F (line 369) investigating the localization of DYS-1 in a *dsh-1* mutant background, and presenting the localization at both extremities as evidence for modification of DYS-1 localization by *dsh-1* loss. I see no difference with figure 3: DYS-1 is always present throughout the muscle cells. This is also in line with the conclusion on line 376 that the interaction with the DPAC is downstream of planar polarization.
7. In figure 6F, the DSH-1 localization is not well visible, I can’t see the restoration of DSH-1 polarity claimed in line 483.

Reviewer #3 (Remarks to the Author):

This is a very well written article describing the function of Wnt signaling components in the establishment and maintenance of polarized protein accumulation in *C. elegans* muscles. In addition, they describe subcellular domains that contain unique combinations of potassium channels. I would not have expected the heterogeneity of distribution for these ion channels a priori. The identification of EGL-20/CAM-1/DSH-1 as regulators of asymmetric cell behavior is not novel, but very little work has been done on the role of these proteins in muscles, nor is there really any prior knowledge of Wnt signaling regulating potassium channel localization. Therefore, the work here is going to be interesting to several fields, and the work highly relevant. The authors also create a number of new mutant alleles, and new fluorescently-tagged strains. These are reagents that will be of use to the community. I have no substantive concerns about the study, it has been rigorous and thorough. The authors have used multiple alleles, tested different knock-in strains, and, have for the most part done a thorough job in analyzing the data. I have one major issue, and a few minor issues that should be addressed before I think this is ready for publication.

Major Issue

- 1) Overall, Figure 5 is very difficult to interpret compared to the earlier figures. The scale is zoomed out, the yellow arrows/carets on the muscles are not as easy to observe as the outlining in the earlier

pictures. For the genes with no phenotype, I would suggest putting those in a supplementary figure. And then to make a more careful presentation of the data in 5A and 5B. It is not clear what the difference in phenotypes is in the anterior versus posterior in Figure 5D. Further, there does not seem to be any quantification that would help the reader understand what differences, if any, were found between the different mutants/alleles, etc. I would require a rather major reorganization of that figure to be as clear as possible about the phenotype and whether the allelic variants have any informative differences.

Minor Issues

1) Dishevelleds frequently display compensation. I think it would be important for the authors to test a double *dsh-1mig-5* mutant. The phenotype in *dsh-1* does have a very strong phenotype, but the TWK-28 proteins are still polarized, being absent from the central region of the muscle.

2) Could the authors indicate in the results where these proteins were tagged? I think that would be important to better understand the outcomes. I believe that they are all c-terminal fusions, but that should be made clear.

3) In Figure 2 the background intensity for TWK-28 is significantly different between wild-type animals and mutants. I would guess that reflects protein that is improperly secreted. Could the authors confirm whether these are single planes or confocal stacks? In the intensity quantification they mention a region of interest that is 5 micrometers high, is that depth of muscle?

4) Could the authors comment on any rationale for testing 4 different *cam-1* alleles, including 2 separate deletions?

Suggestions

1) In lines 141-142 the authors mention some of the proteins exhibit, localization to the lateral or internal surfaces. It might be helpful to provide a coronal cross-section in Figure 1 A to orient the non-C. elegans expert to where those surfaces are in the muscle.

Revision of NCOMMS-23-48419, Peysson *et al.*

We wish to thank the three reviewers for their questions, suggestions, and very positive comments about our study. We hope that this revised manuscript will satisfactorily address the comments that have been raised.

REVIEWER COMMENTS**Reviewer #1 (Remarks to the Author):**

This is a comprehensive and very interesting study. Using the *C. elegans* muscle cells as a model, the authors showed that a subgroup of potassium channels (TWK-28, TWK-24, and SLO) are asymmetrically localized within individual muscle cells, and that this asymmetry is extended across the entire musculature, similar to classical planar cell polarity (PCP) in the fly and vertebrate epithelia. This observation is novel, as such tissue-level planar polarity has not been previously described in *C. elegans*. Using an extended series of genetic analyses, the authors went on to demonstrate that members of the dystrophin-associated protein complex (DAPC) play an important role in membrane expression/retention of the potassium channels, but not in these channels' asymmetrical distribution. Importantly, the classical PCP pathway also appears to be dispensable for asymmetrical channel distribution, and instead the Wnt-Ror-Dvl pathway was found to be required. The study presents an elegant example where the Wnt-Ror-Dvl pathway imparts asymmetry at both the cellular and tissue levels, and that this occurs independently of the classical PCP pathway. It is likely that this functional distinction between the two morphogenetic signaling systems, as revealed by the study, is conserved in other biological contexts, and in other metazoan species. The experimental paradigm described in the study should also provide a powerful tool for future mechanistic investigation of Wnt-Ror-Dvl signaling in symmetry breaking. Overall, I am enthusiastic about the work and believe that it represents a significant advancement in the field.

Suggestions to authors:

1. The phenotypes described in Figs. 5B-D, 5F, and 6E, 6F should be quantified and statistically analyzed. Ideally, statistical analysis should also be done for the *dsh1* effects on asymmetrical localization of TWK-28, SLO-1 and TWK-24 shown in Fig. 4A-C.

We now provide statistical analysis of data presented in Fig. 4C. This analysis shows a significant change in the position of posterior boundaries for each marker, but not of the anterior boundaries.

Regarding the quantification of muscle polarity (based on the distribution of TWK-28), the phenotypes caused by time-resolved degradation/recovery of DSH-1, and the phenotypes caused by loss-of-function mutations of *dsh-1*, *mig-14*, *egl-20* and *cam-1* were fully penetrant, in the sense that we saw multiple defective/non-polarized cells in each animal. Thus, we assessed at least 25 animals for each condition, and observed multiple cells in each animal for the different mutant contexts. Similarly, we considered that a given *dsh-1* cDNA was functionally sufficient to establish/rescue polarity if the *dsh-1* polarity defect was reversed in multiple cells, of multiple transgenic animals. We have updated the materials and methods section to clarify this point.

2. Presentation of Fig. 5 can be improved. The boundaries of individual muscle cells are difficult to discern without the magenta Twk-12-wrmScarlet marker as shown in Fig. 1 and 4. In the legend, the authors stated "Left- or right-pointing yellow arrowheads indicate anterior or posterior extremity of muscle cells, respectively", however, it is still difficult to understand

what these arrowheads are supposed to show, since many panels only have left pointing arrowheads, in the middle of the green twk-28 signals. Showing zoomed-in images of the muscle cells with more clearly marked boundaries would be helpful. It would also be helpful to annotate the green signals as twk-28 (as shown in Fig. 1B and 4A).

We provide an updated Figure 5 in this revised manuscript. As requested, we now indicate that TWK-28 is shown in all panels. Also, as suggested by Reviewer 3, we have moved the panels from Fig 5C to Supplementary Fig. S6, hence removing many instances where muscle cells were not defective and only left-pointing arrowheads were shown.

We agree that outlining cells using a muscle membrane marker is the ideal way to illustrate muscle polarity. For some markers, the outline of the cell is discernable without specific membrane labeling which allowed us to draw cell outlines manually. This is unfortunately not the case for the TWK-28 channel. However, to an experienced observer, the juxtaposition of TWK-28 domains in successive muscle cells (forming a head-to-tail configuration) is a simple readout for muscle polarity, and we relied on this read-out to explore the many mutants which were shown in Figure 5 and Figure S6. To help the reader interpret our images we tested several different representations, before eventually deciding to use arrowheads because it is unobtrusive and provides a visual representation of the characteristic head-to-tail configuration. In the future, we will systematically use membrane markers even though they require additional rounds of genetic crossing and may sometimes even require additional genome engineering steps when loci are genetically linked.

3. In Fig. 5D legend, the authors stated “Anterior mid-body muscle cells in *cam-1* and *egl-20* mutants are wild type”. By “wild type”, do they mean “phenotypically normal”? This should be made more clear.

We thank the reviewer for this comment. We have clarified this figure legend by stating:

“In cam-1 and egl-20 mutants, polarity is not affected in muscle cells situated anteriorly to the vulva (“Anterior midbody”).”

4. There appear to be many more of the very large punctate green twk-28 signals (e.g. Panel D, wild type, anterior midbody, near the left bottom quadrant of the image; Panel D, *dsh-1* posterior midbody, bottom part of the image). Is there an explanation for this, and what might these large puncta be?

The characteristic shape and intensity of this fluorescent signals is typical of intestinal lysosome-related autofluorescence. This non-specific fluorescent signal is generally quite variable and is more or less visible depending on the orientation of the animal. We have added white asterisks to the relevant panels and clarified this point in the figure legend by adding the following sentence:

“Non-specific intestinal lysosome-related autofluorescence is visible in some panels and labelled with a white asterisk.”

5. Fig. 6A, again the use of the arrowheads is confusing. It is also difficult to distinguish the white and yellow arrowheads in the rather small images.

In panel 6A, we chose to provide a “zoomed out” view to illustrate the tissue polarity of DSH-1 localization. In panel 6B, we perform a detailed localization analysis at the cellular level. Finally, in panel 6C, we show additional views for the localization of DSH-1, this time relative to TWK-28. The magnified views in panel 6C, allow us to more clearly illustrate the juxtaposition of the DSH-1 and TWK-28 fluorescent signals, in the posterior and anterior of two muscle cells, respectively. We hope that, taken together, these different levels of detail provide the reader with a comprehensive understanding of DSH-1 distribution.

During the preparation and revision of our figures, we have tested several different representations. In the end, the use of arrowheads appeared to us as the simplest and most

efficient way to orient the reader. Yet, we realize that the arrowheads were too small in our initial version. We have increased the size of the arrowheads and increased their thickness, making them more clearly visible in this revised manuscript. We have also paid attention to provide high resolution images that allow the reader to zoom in on details of interest in each panel.

Reviewer #2 (Remarks to the Author):

In this study, Peysonn et al. investigate the localization of potassium channels and the dystrophin-associated protein complex in muscle cells in the nematode *C. elegans*. Unexpectedly, the authors uncover highly specific subcellular localization patterns for different potassium channels. Moreover, the localization patterns appear to be governed by some type of planar cell polarity mechanism, with channels localizing to the anterior, posterior, or center of muscle cells.

In the next part of the manuscript the authors investigate the requirement of different components of the dystrophin-associated protein complex (DPAC) for localization of the TWK-28 channel, and the subcellular localization of DPAC components.

In the final part of the manuscript the authors investigate the mechanism behind the planar polarization of the channels and the DPAC. They find that the Wnt-Ror-Dvl pathway is necessary for the planar polarization of muscle cells.

The identification of planar polarized localization patterns for ion channels in muscle is both exciting and novel, as to my knowledge muscle cells in *C. elegans* were not previously known to show planar polarity. I found the overall study to be impressively complete and well documented, with extensive investigations into the entire DPAC 'complex', the ion channels, potential PCP pathways, Wnt ligands and other Wnt signaling components. Many new transgenic lines were made to investigate subcellular localization and co-localizations, structure function analyses are presented of DSH-1 and CAM-1, and developmental requirements are investigated using inducible protein degradation.

Given the conserved nature of ion channels, DPAC components, and Wnt signaling, the findings are likely to be relevant to a broad audience beyond the *C. elegans* community. I gladly recommend publication, and have only a few small points that might be addressed.

1. Keeping track of the many components tested for non-muscle specialists would be helped by a schematic overview of the possible DPAC components and their mammalian/worm names.

We now provide a summary table (Supplementary table ST1) listing the names of *C. elegans* DPAC proteins and their corresponding human orthologs.

2. The final sentence of the introduction 'first examples of quintessential planar cell polarity patterns' is overstating the findings. Canonical PCP pathways are not involved in muscle, and planar Wnt polarity is well established in the seam cells. The study is exciting enough without claiming to be the first ever to show PCP in worms.

In this revised manuscript, we have rephrased this sentence in the following way:

"Taken together, these findings uncover the intricate organization of the worm's sarcolemma and reveal a new instance of planar cell polarity in *C. elegans*."

3. In line 217 I was expecting Figure S1A to show the 'no obvious impact', but it only shows the wild-type distribution of channels. I don't think the lack of effect of *dys-1* loss is shown. It would be more appropriate to list this as data not shown.

The first row of panels of Figure S1A corresponds to the wild-type background, while the second row shows representative images in the *dys-1* mutant context.

4. Line 254: analyse dystrophin dynamics. This to me implies looking at protein dynamics (e.g. FRAP), but it is expression/translation dynamics that are followed.

We fully agree. To clarify this point, we have edited this statement in the revised manuscript and now state:

“Using this assay, we could analyse the kinetics of dystrophin recovery over 24 hours, starting at the L4 larval stage (Figure 2F).”

5. On line 333: three Dsh orthologs. Mig-5 and *dsh-1* are investigated but *dsh-2* is left out. Is there a specific reason for not mentioning *dsh-2* here? The rest of the study is so complete, it seems an odd omission.

We initially chose not to test *dsh-2* for two reasons: (1) because *dsh-2* was not expressed in muscle cells to the best of our knowledge and (2) because we had already identified a strong and penetrant phenotype for *dsh-1*. For this revised manuscript, we have now generated a molecular null mutant of *dsh-2* and determined that it does not affect muscle polarity. We have included this new data in Supplementary Figure 6 to provide a more comprehensive analysis.

6. I did not understand the analysis in Figure 4F (line 369) investigating the localization of DYS-1 in a *dsh-1* mutant background, and presenting the localization at both extremities as evidence for modification of DYS-1 localization by *dsh-1* loss. I see no difference with figure 3: DYS-1 is always present throughout the muscle cells. This is also in line with the conclusion on line 376 that the interaction with the DPAC is downstream of planar polarization.

Due to the broad distribution of DYS-1 throughout the sarcolemma, it is indeed more difficult to discern whether DYS-1 accumulates in a polarized distribution at the extremities of muscle cells. We therefore used TWK-28 as a landmark to pinpoint clusters of DYS-1 that are present in muscle extremities.

In the anterior tip, DYS-1/TWK-28-containing clusters form a small line of fluorescent dots in wild-type and *dsh-1(0)* (see figure below). In *dsh-1* mutants, the same linear arrangement of TWK-28 is found in the posterior tip, and one can observe the colocalization of DYS-1 there. For clarity, we provide hereafter a magnified view of the panels presented in Figure 4F.

We realize that an alternative model could be considered in which DYS-1 is present in the posterior but can only recruit TWK-28 in the absence of DSH-1. However, the fact that Sarcoglycans are also present, in a comet-shaped pattern, at the posterior end of muscle cells only in *dsh-(0)* (Figure 4G), makes this model less likely and is in favor of the notion that the dystrophin-associated protein complex, and its core component DYS-1, is also present in a comet-shaped pattern in the posterior in *dsh-1* mutants.

7. In figure 6F, the DSH-1 localization is not well visible, I can't see the restoration of DSH-1 polarity claimed in line 483.

DSH-1 detection is very difficult in muscle cells of "aged" animals. We have modified our statement to clarify that we are evaluating the recovery of muscle polarity based on the localization of TWK-28, and not DSH-1 itself.

"In turn, over the course of the 72-hour recovery period, we observed a restoration of the polarized distribution pattern of TWK-28."

Reviewer #3 (Remarks to the Author):

This is a very well written article describing the function of Wnt signaling components in the establishment and maintenance of polarized protein accumulation in *C. elegans* muscles. In addition, they describe subcellular domains that contain unique combinations of potassium channels. I would not have expected the heterogeneity of distribution for these ion channels a priori. The identification of EGL-20/CAM-1/DSH-1 as regulators of asymmetric cell behavior is not novel, but very little work has been done on the role of these proteins in muscles, nor is there really any prior knowledge of Wnt signaling regulating potassium channel localization. Therefore, the work here is going to be interesting to several fields, and the work highly relevant. The authors also create a number of new mutant alleles, and new fluorescently-tagged strains. These are reagents that will be of use to the community. I have no substantive concerns about the study, it has been rigorous and thorough. The authors have used multiple alleles, tested different knock-in strains, and, have for the most part done a thorough job in analyzing the data. I have one major issue, and a few minor issues that should be addressed before I think this is ready for publication.

Major Issue

1) Overall, Figure 5 is very difficult to interpret compared to the earlier figures. The scale is zoomed out, the yellow arrows/carets on the muscles are not as easy to observe as the outlining in the earlier pictures. For the genes with no phenotype, I would suggest putting those in a supplementary figure. And then to make a more careful presentation of the data in 5A and 5B. It is not clear what the difference in phenotypes is in the anterior versus posterior in Figure 5D. Further, there does not seem to be any quantification that would help the reader understand what differences, if any, were found between the different mutants/alleles, etc. I would require a rather major reorganization of that figure to be as clear as possible about the phenotype and whether the allelic variants have any informative differences.

We agree that outlining cells using a muscle membrane marker is the ideal way to illustrate muscle polarity. For some markers, the outline of the cell is discernable without specific membrane labeling which allowed us to draw cell outlines manually. This is unfortunately not the case for the TWK-28 channel. However, to an experienced observer, the juxtaposition of TWK-28 domains in successive muscle cells (forming a head-to-tail configuration) is a simple read-out for muscle polarity, and we relied on this strategy to explore the many mutants which were shown in figure 5. To help the reader interpret our images we tested several different representations, before eventually deciding to use arrowheads because it was unobtrusively and provides a visual representation of the characteristic head-to-tail configuration. In the future, we will systematically use membrane markers even though they require additional rounds of genetic crossing and may sometimes even require specific genome engineering when loci are genetically linked.

As requested, we have moved the panels from Fig 5C to Supplementary Fig. S6 in this revised manuscript.

Regarding 5D, we have now clarified this point in the figure legend by stating:

"In cam-1 and egl-20 mutants, polarity is not affected in muscle cells situated anteriorly to the vulva ("Anterior midbody")."

Regarding the quantification of muscle polarity (based on the distribution of TWK-28), the phenotypes of *dsh-1*, *mig-14*, *egl-20* and *cam-1* were fully penetrant, in the sense that we saw multiple defective/non-polarized cells in each animal. Thus, we assessed at least 25 animals for each condition, and observed multiple cells in each animal for the different mutant contexts. Similarly, we considered that a given *dsh-1* cDNA was functionally sufficient to establish/rescue polarity if the *dsh-1* polarity defect was reversed in multiple cells, of multiple transgenic animals. We have updated the materials and methods section to clarify this point.

Minor Issues

1) Dishevels frequently display compensation. I think it would be important for the authors to test a double *dsh-1mig-5* mutant. The phenotype in *dsh-1* does have a very strong phenotype, but the TWK-28 proteins are still polarized, being absent from the central region of the muscle.

To address this question, we have now built a double mutant using *mig-5* and *dsh-1* molecular null alleles. The double mutant appears identical to the single *dsh-1(0)* mutant regarding muscle polarity (see images below). TWK-28 remains enriched at both extremities and absent from the central region of the muscle.

2) Could the authors indicate in the results where these proteins were tagged? I think that would be important to better understand the outcomes. I believe that they are all c-terminal fusions, but that should be made clear.

We now provide this information for all knock-in lines in Table S2, which describes each allele used in our study.

3) In Figure 2 the background intensity for TWK-28 is significantly different between wild-type animals and mutants. I would guess that reflects protein that is improperly secreted. Could the authors confirm whether these are single planes or confocal stacks? In the intensity quantification they mention a region of interest that is 5 micrometers high, is that depth of muscle?

The region of interest that was used to quantify fluorescence in each comet was 5 μm wide by 30 μm long (along the antero-posterior axis). In each case, we summed the fluorescence intensity of slices corresponding to a 2.5 μm -deep volume. Fluorescence was then measured by subtracting the background signal present outside of the comet region. The images we show however are the raw image projections, without any form of signal subtraction. This is why some images have a slightly higher background surrounding the comet region.

4) Could the authors comment on any rationale for testing 4 different *cam-1* alleles, including 2 separate deletions?

We tested multiple alleles, first, to firmly and definitively demonstrate the requirement for CAM-1, and second, in the hope of identifying CAM-1 alleles with different effects. However, we saw no obvious difference between the alleles, but decided to nevertheless provide the full information in our manuscript.

Suggestions

1) In lines 141-142 the authors mention some of the proteins exhibit, localization to the lateral or internal surfaces. It might be helpful to provide a coronal cross-section in Figure 1 A to orient the non-*C. elegans* expert to where those surfaces are in the muscle.

Following this reviewer's recommendation, we now include a cross-section clarifying the organization of *C. elegans* musculature in a transverse view.

REVIEWERS' COMMENTS

Reviewer #1 (Remarks to the Author):

This is a revised manuscript by Peysson et al. I was previously enthusiastic about the study and continue to feel the same. The authors have adequately addressed my comments, and I am pleased to recommend the manuscript for publication. My only comment is that I am still a bit confused by the arrow heads in Figures 5 and 6. I am concerned that some of the readers might find it confusing as well. I will elaborate my confusion below, with the hope that the authors will clarify this in the final published version.

Based on the yellow outline of the cell shown in Figure 1C, the magenta labeling of the cell boundary and the green TWK-28 signals shown in Figure 4A and B (left most panels), and the yellow and white dotted lines shown in Figure 6C (the bottom three panels), I would expect that when the authors say that the yellow and white arrow heads in Figures 5 and 6 are meant to indicate “the anterior and posterior extremity of muscle cells”, these arrow heads would be placed at the most anterior and posterior boundaries of the cells, with the side way “V” shape coinciding with the physical boundaries of the cell. The way that the arrow heads are placed, it seems as if the TWK-28 signals are extending past ends of the cells. To demonstrate my point better, I have attached a screenshot of Figure 5A and C in which I added some red arrow heads to demarcate where I would imagine the “extremity” of the cells to be (akin to the dotted lines in Figure 6C, bottom three panels). I have further marked two of the red arrow heads in Figure 5C with the numbers 1 and 2 (for the purpose of this discussion; not meant to be shown in the actual paper) wonder if these are in the same or different cells. Lastly, in Figure 5, both the forward and backward arrow heads are currently in yellow, but in Figure 6, the forward and back arrow heads are in yellow and white, respectively. They should make this more consistent between the two figures. In my opinion, making all of them yellow and explaining what they are meant to indicate, as they have done in Figure 5 legend, is sufficient. White and yellow are actually a bit difficult to distinguish.

Reviewer #2 (Remarks to the Author):

I want to thank the reviewers for the clarifications and changes to the manuscript they have made in response to my minor points. They have all been addressed to my full satisfaction.

Reviewer #3 (Remarks to the Author):

The authors have satisfactorily addressed all of the concerns that I had with the manuscript. I now recommend this manuscript for publication.

Figure 5

A

DSH-1a

+ *Pmyo-3::dsh-1a* Δ DIX

B

C

Anterior midbody

Wild type

dsh-1(ok1445)

Revision of NCOMMS-23-48419B, Peysson *et al.*

We wish to thank the three reviewers for providing their assessment of our revised manuscript. We hope this new version will satisfactorily address the remaining comments.

Reviewer #1 (Remarks to the Author):

This is a revised manuscript by Peysson *et al.* I was previously enthusiastic about the study and continue to feel the same. The authors have adequately addressed my comments, and I am pleased to recommend the manuscript for publication. My only comment is that I am still a bit confused by the arrow heads in Figures 5 and 6. I am concerned that some of the readers might find it confusing as well. I will elaborate my confusion below, with the hope that the authors will clarify this in the final published version.

Based on the yellow outline of the cell shown in Figure 1C, the magenta labeling of the cell boundary and the green TWK-28 signals shown in Figure 4A and B (left most panels), and the yellow and white dotted lines shown in Figure 6C (the bottom three panels), I would expect that when the authors say that the yellow and white arrow heads in Figures 5 and 6 are meant to indicate “the anterior and posterior extremity of muscle cells”, these arrow heads would be placed at the most anterior and posterior boundaries of the cells, with the side way “V” shape coinciding with the physical boundaries of the cell. The way that the arrow heads are placed, it seems as if the TWK-28 signals are extending past ends of the cells. To demonstrate my point better, I have attached a screenshot of Figure 5A and C in which I added some red arrow heads to demarcate where I would imagine the “extremity” of the cells to be (akin to the dotted lines in Figure 6C, bottom three panels). I have further marked two of the red arrow heads in Figure 5C with the numbers 1 and 2 (for the purpose of this discussion; not meant to be shown in the actual paper) wonder if these are in the same or different cells. Lastly, in Figure 5, both the forward and backward arrow heads are currently in yellow, but in Figure 6, the forward and back arrow heads are in yellow and white, respectively. They should make this more consistent between the two figures. In my opinion, making all of them yellow and explaining what they are meant to indicate, as they have done in Figure 5 legend, is sufficient. White and yellow are actually a bit difficult to distinguish.

We have modified our manuscript figures following the reviewer’s requests/comment. Spécifically,

a) “[...] these arrow heads would be placed at the most anterior and posterior boundaries of the cells [...]”

We have modified our figures according to the reviewer’s suggestion.

b) “I have further marked two of the red arrow heads in Figure 5C with the numbers 1 and 2 (for the purpose of this discussion; not meant to be shown in the actual paper) wonder if these are in the same or different cells.”

Indeed, the red arrow heads correspond to two different cells.

c) “Lastly, in Figure 5, both the forward and backward arrow heads are currently in yellow, but in Figure 6, the forward and back arrow heads are in yellow and white, respectively.”

In the updated Figure 6, we have colored all arrowheads in yellow, as per the reviewer’s request.

Reviewer #2 (Remarks to the Author):

I want to thank the reviewers for the clarifications and changes to the manuscript they have made in response to my minor points. They have all been addressed to my full satisfaction.

Reviewer #3 (Remarks to the Author):

The authors have satisfactorily addressed all of the concerns that I had with the manuscript. I now recommend this manuscript for publication.